# Mitochondrial dynamics controls anti-tumour innate immunity by regulating CHIP-IRF1 axis stability

Zhengjun Gao[1], Yiyuan Li[1], Fei Wang[1], Tao Huang[1], Keqi Fan[1], Yu Zhang[2], Jiangyan Zhong[1], Qian Cao[2], Tong Chao[1], Junling Jia[1], Shuo Yang[3,4], Long Zhang [1], Yichuan Xiao[5], Ji-Yong Zhou[6], Xin-Hua Feng[1] & Jin Jin[1,6]

Macrophages, dendritic cells and other innate immune cells are involved in inflammation and host defense against infection. Metabolic shifts in mitochondrial dynamics may be involved in Toll-like receptor agonist-mediated inflammatory responses and immune cell polarization. However, whether the mitochondrial morphology in myeloid immune cells affects anti-tumor immunity is unclear. Here we show that FAM73b, a mitochondrial outer membrane protein, has a pivotal function in Toll-like receptor-regulated mitochondrial morphology switching from fusion to fission. Switching to mitochondrial fission via ablation of *Fam73b* (also known as *Miga2*) promotes IL-12 production. In tumor-associated macrophages, this switch results in T-cell activation and enhances anti-tumor immunity. We also show that the mitochondrial morphology affects Parkin expression and its recruitment to mitochondria. Parkin controls the stability of the downstream CHIP–IRF1 axis through proteolysis. Our findings identify mechanisms associated with mitochondrial dynamics that control anti-tumor immune responses and that are potential targets for cancer immunotherapy.

[1] Life Sciences Institute, Zhejiang University, Hangzhou 310058, China. [2] Sir Run Run Shaw Hospital, College of Medicine Zhejiang University, Hangzhou 310016, China. [3] Department of Immunology, Nanjing Medical University, Nanjing, 211166 Jiangsu Province, China. [4] State Key Laboratory of Reproductive Medicine, Nanjing Medical University, Nanjing, 211166 Jiangsu Province, China. [5] Institute of Health Sciences, Shanghai Institutes for Biological Sciences, Chinese Academy of Sciences/Shanghai Jiao Tong University School of Medicine, Shanghai 200031, China. [6] Key Laboratory of Animal Virology of Ministry of Agriculture, Zhejiang University, Hangzhou 310058, China. Zhengjun Gao and Yiyuan Li contributed equally to this work. Correspondence and requests for materials should be addressed to J.J. (email: jjin4@zju.edu.cn)

Mitochondria generate cellular energy that can alter their morphology through biogenesis and the opposing processes of fission and fusion[1, 2]. Maintenance of dynamic mitochondrial networks has crucial effects on signaling transduction, ATP production, iron–sulfur cluster biogenesis, and calcium buffering[3, 4]. Mitochondrial morphology switching is also implicated in cell survival, apoptosis, and cellular metabolic homeostasis[5, 6]. Both the outer membrane (OM) and inner membrane (IM) are involved in mitochondrial fusion and fission. Mitofusin 1 (MFN1) and MFN2 are essential for OM fusion and maintenance of mitochondrial morphology[7, 8]. Optic atrophy 1 (OPA1) controls IM fusion and protects cells with mitochondrial dysfunction due to *Mfn1* deficiencies[9–11]. Two OM proteins, named FAM73a (also known as mitoguardin 1) and FAM73b, are required for mitochondrial fusion, and they function by regulating phospholipid metabolism via mitochondrial

phospholipase D (MitoPLD)[12]. Although the physiological functions of mitochondria are linked to their morphology[13], mitochondrial dynamics in immune responses are not clear owing to the embryonic lethality of MFN1/2 double knockout (KO) or OPA1 mutant mice. However, FAM73a and FAM73b KO mice are viable and exhibit only moderately decreased body weight and body fat. Therefore, FAM73a and FAM73b KO mice are suitable models to evaluate the role of mitochondrial dynamics in immune homeostasis and host defense.

Mitochondria have essential functions in both innate and adaptive immunity. Mitochondria are catabolic organelles and are the major source of cellular ATP and ROS, which are important in innate immune responses to cellular damage, stress, and infection[14–16]. Mitochondria also host signaling modulators such as mitochondrial antiviral signaling protein (MAVS) and evolutionarily conserved signaling intermediate in Toll pathway,

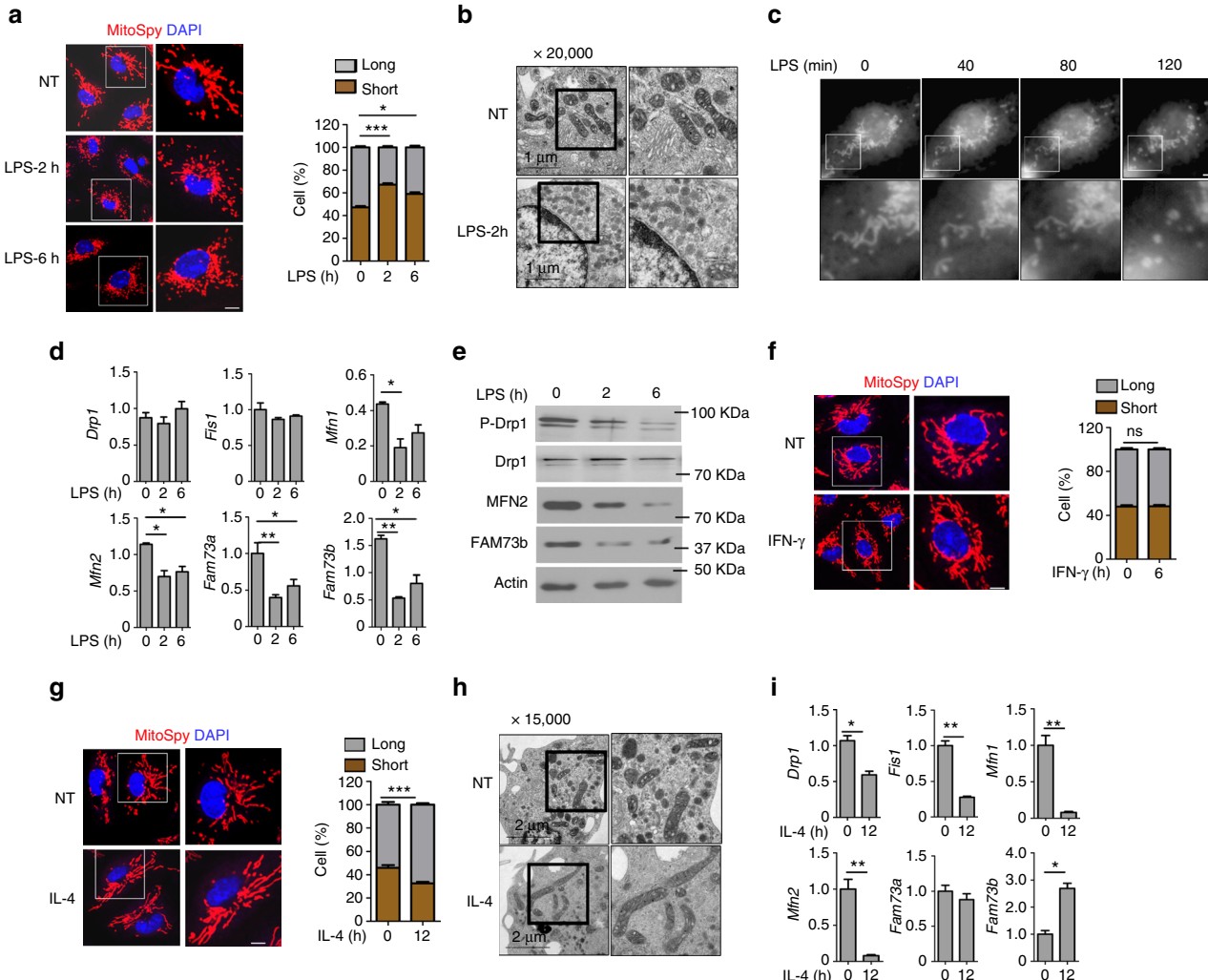

**Fig. 1** FAM73b controls mitochondrial morphology. Primary BMDMs treated with LPS (500 ng/ml) at the indicated time points. Mitochondria were visualized using MitoSpyTM Orange CMTMRos staining. Representative confocal images of the mitochondrial morphology are shown, and quantification was analyzed with Image-Pro and presented as a bar graph (**a**). The data are shown as the mean ± SEM of three independent experiments, with 100 cells counted for each replicate; colors indicate the mitochondrial morphology (long or short). **b** The mitochondrial morphology of WT BMDMs was analyzed with EM (scale bar, 1 μm). **c** Time-lapse images of WT BMDMs treated with LPS (500 ng/ml) and stained with MitoSpyTM Orange CMTMRos. **d**, **e** qRT-PCR and IB analyses of the indicated genes using LPS-stimulated BMDMs from WT mice. **f** Representative confocal images and bar graph of WT BMDMs stimulated with IFN-γ (10 ng/ml) for 12 h. (**g**, **h**) WT BMDMs stimulated with IL-4 (20 ng/ml) for 12 h. Representative confocal (**g**) and EM (**h**) images are presented. **i** The indicated genes were measured with qRT-PCR. All qRT-PCR data are presented as fold-induction relative to the *Actb* mRNA level. All the data are representative of three independent experiments. Scale bars in **a**, **c**, **f** and **g** are 5 μm; one in **b** is 1 μm; one in **h** is 2 μm. Error bars are the mean ± SEM values. Two-tailed unpaired t-tests were performed. *P < 0.05; **P < 0.01

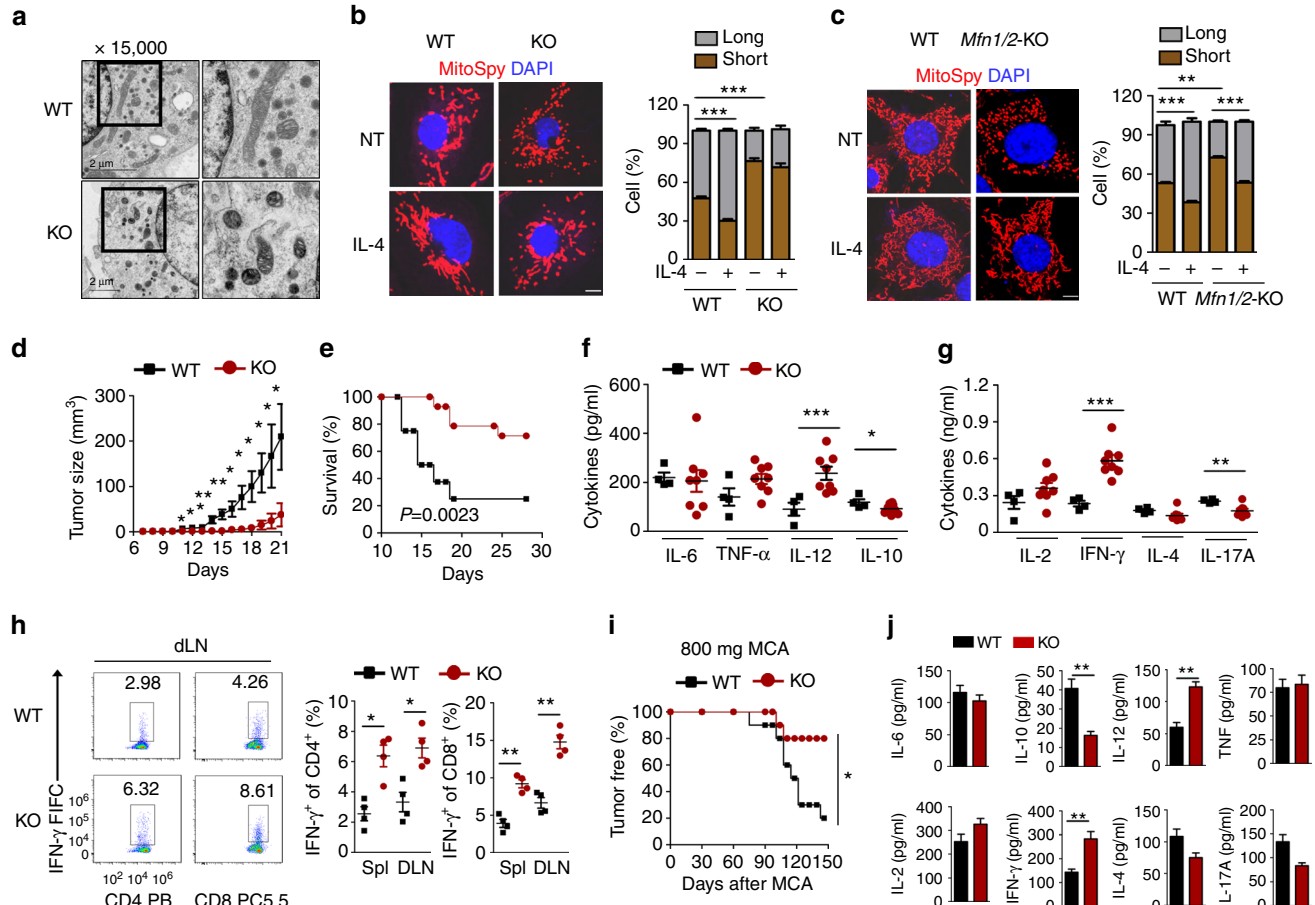

**Fig. 2** FAM73b-deficient mice are resistant to tumor growth. **a** Electron microscopy of the mitochondrial morphology from WT and FAM73b KO BMDMs. **b**, **c** Representative confocal images and statistical analysis of the mitochondrial morphology from FAM73b KO BMDMs (**b**) or MFN1/2 KO MEFs (**c**). WT or FAM73b KO (KO) mice (n = 14) were injected with 2 × 10^5 B16 melanoma cells. Tumor growth curve (**d**) and survival curve (**e**). ELISA assay to determine the various cytokine levels from innate (**f**) and adaptive (**g**) immunity in serum on day 15. **h** ICS and flow cytometric analysis of IFNγ⁺ CD4⁺ and CD8⁺ T cells as described above, presented as a representative plot (left panel) and a summary graph (right panel). The percentage is based on total CD4⁺ or CD8⁺ T cells. **i**, **j** WT or FAM73b KO mice (n = 10) were injected with 800 μg of MCA. Frequency of tumor-free mice (**i**). The indicated cytokines were detected in the serum of WT or FAM73b KO mice via ELISA. The serum was collected from the mice on day 150 after injection of MCA (**j**). All the data are representative of three independent experiments. Scale bar in **a** is 2 μm; those in **b** and **c** are 5 μm. Error bars are mean ± SEM values. Two-tailed unpaired t-tests were performed. *P < 0.05; **P < 0.01

mitochondrial (ECSIT) to control pattern recognition receptor (PRR)-mediated type I interferon induction and inflammatory responses[17–22]. Additionally, mitochondria-mediated metabolic changes are associated with immune cell polarization, particularly lymphocyte homeostasis and memory T-cell generation[23]. T-cell differentiation to T helper type 1 (Th1), Th2, and Th17 subpopulations preferentially utilizes glycolysis rather than mitochondrial OXPHOS[24, 25], and T regulatory (Treg) cells have distinct metabolic demands, which are dependent on both lipid metabolism and OXPHOS[24, 25]. Polarization of macrophages also involves different metabolic pathways, with aerobic glycolysis important for M1 macrophages and fatty acid oxidation (FAO)-driven mitochondrial oxidative phosphorylation important for differentiation of M2 macrophages[26, 27].

IL-12 family cytokines are mainly produced by myeloid cells, and they control adaptive immune responses, especially T-cell differentiation[28]. IL-12 p35, IL-12 p40, and IL-23 p19 are proinflammatory cytokines produced by dendritic cells, macrophages and fibroblasts in response to microbial pathogens and tumors[29, 30]. IL-12 and IL-23 expression is associated with epigenetic modifications[31] and various transcription factors, such as c-Rel, IRF5, and IRF1[30]. Genetic evidence indicates that LPS-induced IL-12 p35 expression is reduced in Irf1^{−/−} macrophages. In vitro data show that monoubiquitinated C terminus of HSC70-interacting protein (CHIP; also known as STUB1) is required for proteolytic degradation of IRF1[32]. Furthermore, CHIP is associated with Parkin, which mediates selective autophagy of depolarized mitochondria[33, 34]. Therefore, here, we investigate the effect of mitochondrial membrane remodeling on IL-12 production and the activity of the CHIP–IRF1 axis in anti-tumor immune responses.

Here we characterize a proteolysis-dependent mechanism of inflammatory cytokine production regulated by mitochondrial fission. Genetic ablation of the fusion mediator Fam73b in macrophages and dendritic cells promotes TLR-induced IL-12 expression and inhibits IL-10 and IL-23 expression. Macrophage-derived IL-12 promotes anti-tumor T-cell responses in vivo in mouse melanoma and MCA-induced fibrosarcoma models. Myeloid cell but not T cell conditional knockout mice have enhanced Th1 responses. Fam73b or Mfn1/Mfn2 depletion causes severe mitochondrial fragmentation and degrades monoubiquitinated CHIP. Furthermore, mitochondrial fission promotes accumulation and recruitment of Parkin, which directly induces monoubiquitinated CHIP degradation and stabilizes the

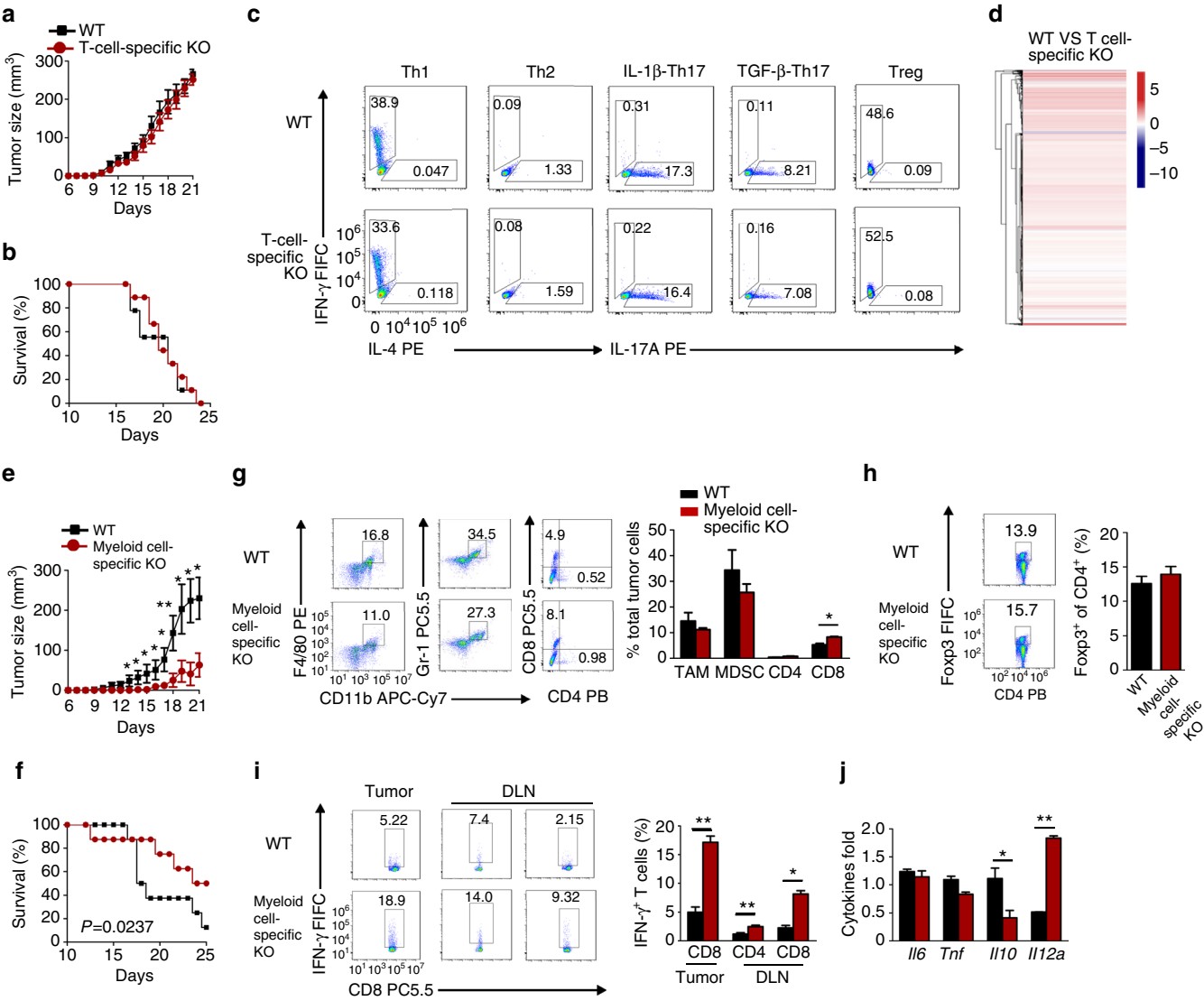

**Fig. 3** FAM73b regulates anti-tumor responses in myeloid cells. **a**, **b** Tumor growth curve and survival curve of T-cell-specific KO mice ($n = 8$) injected with $2 \times 10^5$ B16 melanoma cells. **c** WT and T-cell-specific KO naive CD4$^+$ T cells (CD44$^{lo}$CD62L$^{hi}$) were stimulated for 4 days with plate-bound anti-CD3 and anti-CD28 antibodies under subpopulation conditions. The frequency of T-cell differentiation was analyzed by flow cytometry based on intracellular staining of cytokines. **d** Heat map showing basal differentially expressed genes in naive T cells from WT and T-cell-specific KO mice. **e**, **f** Tumor growth curve (**e**) and survival curve (**f**) of myeloid cell-specific KO ($n = 8$) mice injected with $2 \times 10^5$ B16 melanoma cells. **g**, **h** Flow cytometric analysis of immune cells (CD45$^+$) that infiltrated day 15 tumors and the frequency of CD4$^+$ T cells, CD8$^+$ T cells, macrophages (CD11b$^+$F4/80$^+$), myeloid derived suppressor cells (MDSCs) (CD11b$^+$Gr-1$^+$) (**g**) and Treg cells (CD4$^+$Foxp3$^+$) (**h**). Summary of flow cytometry data based on multiple animals, showing the frequency of the indicated cell populations within total tumor cells (right panel). **i** ICS and flow cytometric analysis of IFNγ$^+$ in CD8$^+$ T tumor or dLN cells, presented as a representative plot (left panel) and a summary graph (right panel). **j** qRT-PCR analysis of the indicated genes in TAMs isolated from tumors in WT or myeloid cell-specific KO mice. All qRT-PCR data are presented as fold-induction relative to the *Actb* mRNA level. The data are the mean ± SEM value of multiple animals (each circle or square represents a mouse) and representative of three independent experiments. Two-tailed unpaired *t*-tests were performed. *$P < 0.05$; **$P < 0.01$

crucial downstream transcription factor IRF1. Our data highlight an unappreciated role of mitochondrial morphology in macrophage polarization and identify an associated signal transduction network.

## Results

**Mitochondrial dynamics involved in macrophage polarization.** To evaluate whether mitochondrial dynamics are involved in macrophage polarization, we stimulated wild-type (WT) bone marrow-derived macrophages (BMDMs) with the TLR4 ligand lipopolysaccharide (LPS) and examined the mitochondrial morphology. Confocal microscopy revealed that LPS-treated

macrophages rapidly and predominantly exhibited punctate mitochondria (Fig. 1a). The intensity of mitochondrial fragmentation was dependent on LPS concentration (Supplementary Fig. 1a). Additionally, the mitochondrial network maintained fission status until 12 h after stimulation (Supplementary Fig. 1b). Ultrastructural analysis using electron microscopy (EM) also indicated that LPS treatment led to small, diverse mitochondria dispersed throughout the cytoplasm (Fig. 1b). Morphometric analysis revealed significantly more mitochondria that occupied a comparable portion of the cellular area (Supplementary Fig. 1c, d). Time-lapse microscopy also showed that mitochondria quickly switched to the fission stage within 2 h (Fig. 1c). We further examined the expression levels of several critical

regulators of mitochondrial dynamics[12, 35]. We found that the canonical fusion mediators *Mfn1/Mfn2* and *Fam73b* were suppressed by LPS stimulation, with reduced phosphorylation of the fission factor Drp1 (Fig. 1d, e). As reported, mitochondrial fission reduces $Ca^{2+}$ uptake and intramitochondrial $Ca^{2+}$ diffusion[36]. Cytosolic $Ca^{2+}$ rise activates the cytosolic phosphatase calcineurin that normally interacts with Drp1. Calcineurin-dependent dephosphorylation of Drp1 regulates its translocation to mitochondria[37]. Similar results were obtained when the TLR3 ligand poly(I:C) was used to stimulate cells (Supplementary Fig. 1b, e). M1 macrophage differentiation is enhanced by interferon-γ (IFN-γ) treatment[38]. In contrast to TLR agonists, IFN-γ alone did not induce extensive fragmentation of mitochondria in BMDMs (Fig. 1f).

IL-4 is known to promote M2 polarization, which is involved in tissue repair and resistance to parasitic helminthes[38]. Recent research has demonstrated that M2 macrophages enhance the mitochondrial oxygen consumption rate (OCR) and spare respiratory capacity (SRC)[27], which suggests that IL-4 might promote mitochondrial fusion. Thus, we treated BMDMs with IL-4 for 12 h and assessed themitochondrial morphology. The data revealed that the mitochondria in M2 macrophages formed elongated tubules, which occupied more cytoplasmic area (Fig. 1g, h, Supplementary Fig. 1c, d). Furthermore, only *Fam73b* levels were increased compared to other genes, such as *Mfn1* and *Mfn2* (Fig. 1i, Supplementary Fig. 1f). These results suggest a potentially essential role of *Fam73b* in mediating mitochondrial morphology during macrophage polarization.

**FAM73b negatively regulates anti-tumor innate immunity**. To study the in vivo function of FAM73b in the immune system, we employed a gene targeting approach (KO first) to delete the *Fam73b* gene in mice (Supplementary Fig. 2a). Immunoblot (IB) analysis revealed loss of FAM73b expression in different types of immune cells in germline FAM73b KO mice (Supplementary Fig. 2b). FAM73b KO mice did not show significant abnormalities in thymocyte development, although they had a moderate increase in the frequency of peripheral CD8$^+$ T cells and all memory T-cell populations (Supplementary Fig. 3a, b). The percentage of Treg cells was comparable both in the spleen and inguinal lymph node (iLN) (Supplementary Fig. 3c). Additionally, deletion of *Fam73b* had no significant effect on the frequency of macrophages, neutrophils or dendritic cells in the bone marrow (BM) and spleen (Spl) (Supplementary Fig. 3d).

Although macrophage development was normal, FAM73b deficiency also triggered a high degree of mitochondrial fragmentation as other cell types[12]. This fission stage could not be rescued by IL-4 treatment as the WT control (Fig. 2a, b). FAM73b KO BMDMs also contained more mitochondria with a comparable occupied area (Supplementary Fig. 1c, d). Additionally, FAM73b deficiency did not affect mitochondrial biogenesis, and cells had a similar mtDNA level (Supplementary Fig. 1e). Interestingly, IL-4 still promoted mitochondrial fusion in $Mfn1/2^{-/-}$ mouse embryonic fibroblasts (MEFs) (Fig. 2c). This result suggests that *Mfns* are dispensable for the mitochondrial morphology switch under polarization stress. Collectively, these data indicate that *Fam73b* functions as a crucial regulator of mitochondrial dynamics during macrophage polarization.

To further investigate the function of FAM73b in regulating the in vivo immune response, we employed a murine melanoma model in which B16 melanoma cells were inoculated into mice. *Fam73b* deletion profoundly suppressed tumor growth and increased the survival rate of the tumor-bearing mice (Fig. 2d, e). Consistently, FAM73b KO mice had enhanced IL-12 and reduced IL-10 serum levels (Fig. 2f). Moreover, the T-cell-derived

cytokine IFN-γ was also upregulated (Fig. 2g). FACS analysis revealed an increased frequency of IFN-γ-positive CD4$^+$ and CD8$^+$ effector T cells both in the draining lymph node (dLN) and the spleen of tumor-bearing FAM73b KO mice (Fig. 2h).

Because the development of primary tumors involves long term interactions between the immune system and tumors, it still remains unknown how the excessive and enhanced production of IL-12 induced by FAM73b deficiency impacts the development of primary tumors. Therefore, we employed a well-characterized methylcholantrene (MCA)-induced fibrosarcoma model. We treated the WT and FAM73b KO mice with 800 μg of MCA and monitored tumor formation for up to 150 days. FAM73b KO mice clearly developed fibrosarcoma at a significantly lower rate of incidence (Fig. 2i). FAM73b KO mouse serum also displayed a profound increase in IL-12 and IFN-γ serum levels (Fig. 2j).

**FAM73b functions in myeloid cells to regulate tumor growth**. To delineate the roles of FAM73b in various immune cells, we bred *Fam73b*-flox mice with mice expressing Cre recombinase driven by the *Cd4* promoter or *lyz2* promoter to generate T-cell- or myeloid cell-specific KO mice. Conditional KO mice exhibited a significant FAM73b deficiency in T cells (Supplementary Fig. 1g) or macrophages (Supplementary Fig. 1h). In the B16 melanoma model, T-cell-specific KO mice had similar tumor growth and death ratio to those of WT mice (Fig. 3a, b). Consistently, we found that T-cell-specific *Fam73b* ablation had no effect on T-cell subtype differentiation in vitro (Fig. 3c). RNAseq analysis also demonstrated that FAM73b-deficient naive CD4$^+$ T cells exhibited gene profiles similar to those of WT cells (Fig. 3d, Supplementary Data 1).

Macrophages are the dominant leukocyte population in the inflammatory microenvironment of tumors[39]. IL-4 from the tumor microenvironment induces cathepsin protease activity in tumor-associated macrophages (TAMs)[40]. To clarify the role of FAM73b in TAMs, we first evaluated the effect of the tumor microenvironment on the mitochondrial morphology. As shown in Supplementary Fig. 4a, b, FAM73b promoted mitochondria switching to the fusion stage when co-cultured with B16 melanoma cells. This fusion was restored by IL-4 neutralizing antibody. In contrast to T-cell-specific KO mice, myeloid cell-KO mice suppressed tumor growth and maintained the survival rate (Fig. 3e, f). Compared to WT hosts, deficiency of FAM73b in myeloid cells led to an increased frequency of infiltrating CD8$^+$ T cells in the tumors, together with a reduction in TAMs and myeloid derived suppressor cells (MDSCs) (Fig. 3g). However, enhanced CD8$^+$ T-cell recruitment was not due to a reduction of Treg cells (Fig. 3h). The tumors and draining lymph node (dLN) of myeloid cell-KO mice contained an increased frequency of IFN-γ-positive CD4$^+$ and CD8$^+$ effector T cells (Fig. 3i). qRT-PCR assays revealed an increase in *Il12a* levels specifically in tumor-infiltrating macrophages from myeloid cell-KO mice (Fig. 3j). Similar to macrophages, FAM73b deficiency in dendritic cells also led to tumor growth resistance, coupled with an increased survival rate (Supplementary Fig. 3c, d). Collectively, these results support an unexpected role for mitochondrial fission in innate cell-mediated anti-tumor immunity.

**Mitochondrial fission promotes TLR-mediated IL-12 induction**. To elucidate the molecular mechanism by which *Fam73b* regulates innate immunity, we analyzed the majority of the transcriptome changes in mRNA abundance. As shown in Fig. 4a and Supplementary Data 2, we identified 786 differentially expressed genes between WT and FAM73b KO macrophages with LPS stimulation (>2 fold). Without stimulation, different genes were mainly enriched in regulating cell survival and

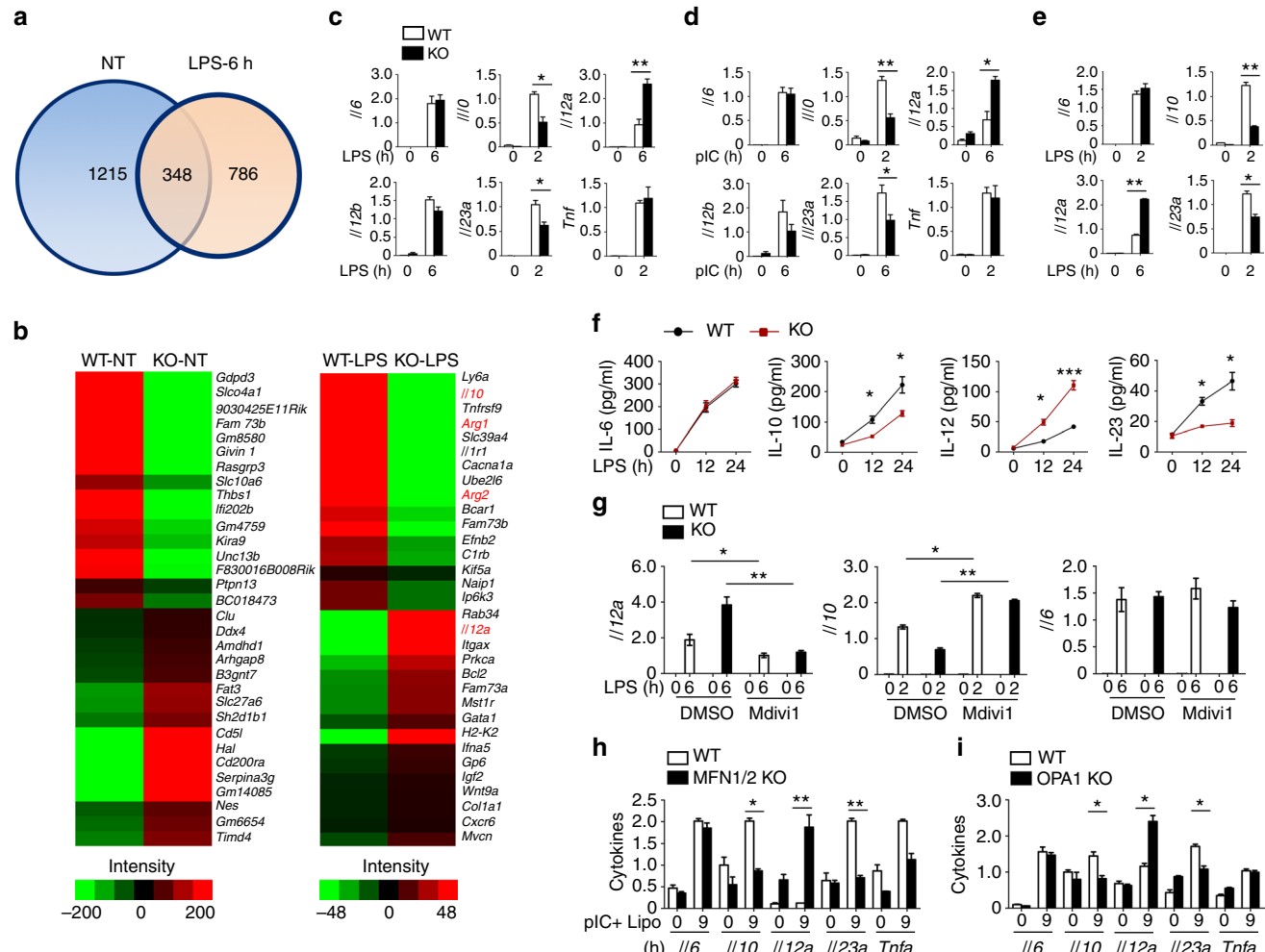

**Fig. 4** Mitochondrial fission is required for induction of *Il12a*. Transcriptome analyses of FAM73b-regulated genes in primary BMDMs from sex-matched WT and FAM73b KO mice (8 weeks old). **a** Venn diagram illustrating the overlap of differentially expressed genes between WT and FAM73b KO BMDMs under non-treatment or LPS-stimulated conditions for 6 h. **b** Heat map showing basal (left panel) and LPS-response (right panel) differentially expressed genes in WT and FAM73b KO BMDMs. qRT-PCR analysis of the indicated genes using WT or FAM73b KO BMDMs stimulated with LPS (**c**) or poly(I:C) (**d**). **e** ELISAs of the indicated cytokines in the supernatants of WT or FAM73b KO BMDMs stimulated with LPS for 12 and 24 h. **f** The indicated gene levels in WT and FAM73b KO BMDCs were measured with qRT-PCR. **g** qRT-PCR analysis of LPS-induced genes changes after treatment with Mdivi-1 for 12 h. **h, i** WT and MFN1/2- (**h**) or OPA1-deficient (**i**) MEFs were stimulated with Lipofectamine-delivered poly I:C for 9 h. The indicated genes were detected by qRT-PCR. All the data are presented as fold-induction relative to the *Actb* mRNA level. The data are presented as the mean ± SEM values and representative of at least three independent experiments. Statistical analyses represent variations in experimental replicates. Two-tailed unpaired *t*-tests were performed. *$P < 0.05$; **$P < 0.01$; ***$P < 0.005$

development (Fig. 4b and Supplementary Fig. 5a). Furthermore, FAM73b deficiency significantly enhanced *Il12a* induction and severely suppressed *Il10* and *Arg1* when responding to TLR stimulation (Fig. 4b and Supplementary Fig. 5b). Cytokine induction was also confirmed by qRT-PCR analysis (Fig. 4c). Additionally, FAM73b appeared to be a common regulator of *Il12a*, because FAM73b also suppressed *Il12a* induction by the polyI:C (TLR3 ligand), R848 (TLR7 ligand), and CpG (TLR9 ligand) (Fig. 4d and Supplementary Fig. 6a, b). Similar results were obtained with ELISAs that detected the secreted cytokines (Fig. 4e). In contrast, FAM73b is required for production of specific markers in M2 polarization (Supplementary Fig. 6c). As seen in macrophages, FAM73b deficiency in bone marrow-derived dendritic cells (BMDCs) also promoted *Il12a* induction (Fig. 4f). A previous report claimed that FAM73a was also required for mitochondrial fusion[12]. Just as with FAM73b, FAM73a KO macrophages produced more IL-12 than macrophages from their WT littermates (Supplementary Fig. 6d, e),

suggesting that FAM73a and FAM73b heterotypic complexes promote mitochondrial fusion and negatively regulate *Il12a* induction.

To further test whether mitochondrial fission is involved in *Il12a* enhancement, WT, and FAM73b KO macrophages were cultured with the fission inhibitor Mdivi-1 before LPS treatment. This drug significantly promoted mitochondrial fusion both in WT and *Fam73b* macrophages (Supplementary Fig. 6f). Mdivi-1 also erased differences in *Il12a* and *Il10* production (Fig. 4g). We further evaluated the function of other fusion-related molecules, such as *Mfn1/Mfn2* and *Opa1*. Due to the embryonic death of these global KO mice, we assessed the cytokine induction in *Mfn1/Mfn2*- or *Opa1*-deleted MEFs. The results indicated that loss of *Mfn1/Mfn2* or *Opa1* also accelerated *Il12a* transcription when MEFs were transfected with Lipofectamine-packaged poly (I:C) (Fig. 4h, i). These results demonstrate an unexpected and pivotal role of mitochondrial dynamics in regulating IL-12 production.

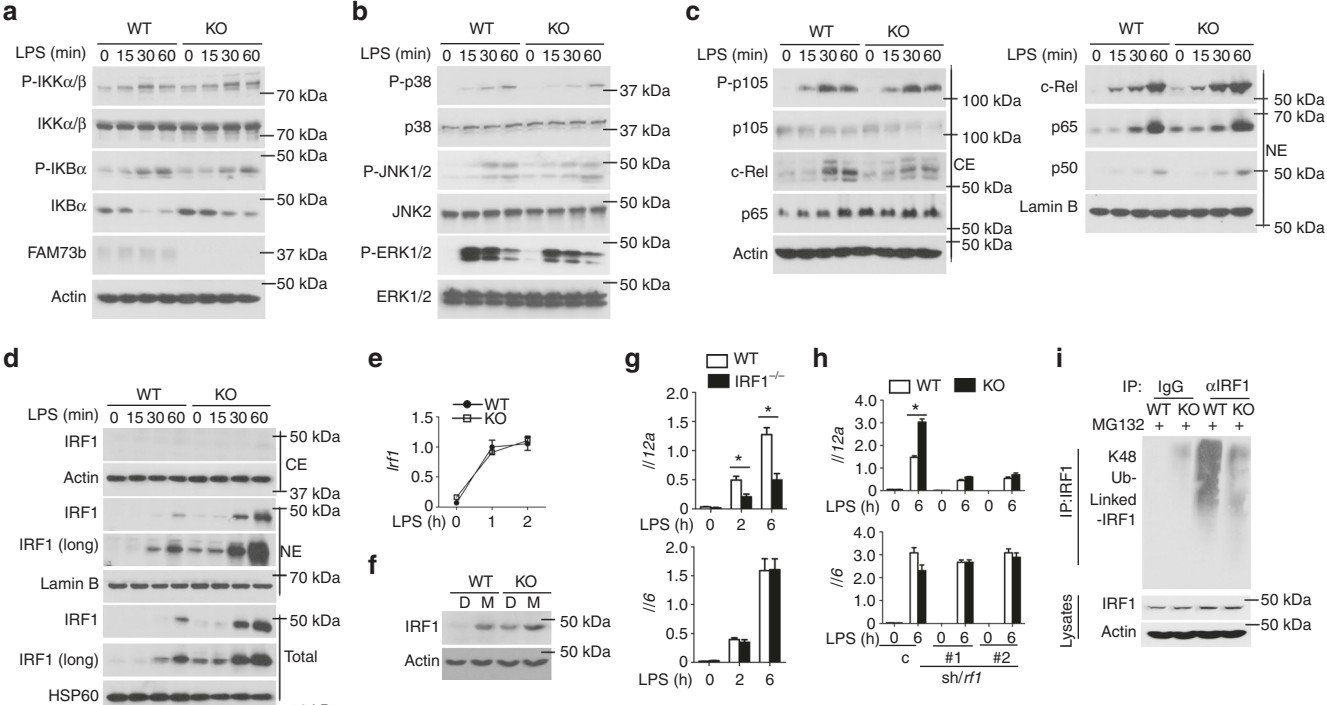

**Fig. 5** FAM73b deficiency promotes IRF1 stability. **a–c** The indicated proteins in whole-cell lysates (**a**, **b**) or cytoplasmic (CE) and nuclear (NE) extracts (**c**) of WT and FAM73b KO BMDMs were measured by IB analysis. **d** IB analysis of the IRF1 level in LPS-stimulated WT or FAM73b KO BMDMs. qRT-PCR analysis of *Irf1* in WT or FAM73b KO BMDMs (**e**). IB analysis of the IRF1 levels in BMDMs after pretreatment with DMSO or MG132 for 30 min before harvest (**f**). **g** *Il12a* and *Il6* mRNA levels were evaluated by qRT-PCR. **h** WT and FAM73b KO BMDMs infected with pGIPZ lentiviral vectors encoding a non-silencing control shRNA (C) or two different *Irf1*-specific shRNAs. qRT-PCR analysis of the indicated genes. The data are presented as the fold-induction relative to the *Actb* mRNA level. **i** WT and FAM73b KO BMDMs were stimulated with LPS for 60 min and incubated with MG132 for 30 min. IRF1 was isolated by IP (under denaturing conditions), and its ubiquitinated form was detected by IB using anti-K48-linked ubiquitin antibody. Protein lysates were also subjected to direct IB (bottom panels). Similar results were obtained in three independent experiments. Two-tailed unpaired t-tests were performed; *$P < 0.05$

**Fam73b deficiency promotes IRF1 accumulation.** Previous studies revealed that BMDCs from *Stat1*$^{-/-}$ and *Ifnar*$^{-/-}$ mice exhibit strongly reduced IL-12 p70 but not IL-12 p40 secretion[41]. Thus, *Fam73b* ablation may improve type I IFN induction to promote *Il12a* expression. In contrast, *Fam73b* deficiency caused a significant reduction in inducible *Ifnb* expression by various stimulators (Supplementary Fig. 7a), without affecting TBK1 and IRF3 phosphorylation (Supplementary Fig. 7b). The results indicated that *Ifnb* may not be the reason for the enhanced *Il12a* levels. Mitochondrial fission-related ROS production generally acts as a signal to trigger pro-inflammatory cytokines[42]. To further assess the role of ROS, we first measured ROS production using FACS. The results revealed that *Fam73b* ablation impaired ROS induction (Supplementary Fig. 7c). Furthermore, the increase in *Il12a* was not restored when cells were incubated with hydrogen peroxide (H$_2$O$_2$) (Supplementary Fig. 7d). Recently, several groups found that mitochondria-mediated metabolic change was involved in establishing immune cell phenotype[26]. Therefore, we compared oxygen consumption by WT and FAM73b KO BMDMs using extracellular flux analysis. FAM73b KO cells exhibited severely decreased OXPHOS activity at the basal level, but the difference was impaired when cells were treated with LPS (Supplementary Fig. 7f), while the relative extracellular acidification rate (ECAR) values were substantially similar between WT and FAM73b KO cells (Supplementary Fig. 7g). WT and FAM73b KO macrophages did not respond to either the FAO inhibitor etomoxir (ETO) or the protonophore uncoupler FCCP (Supplementary Fig. 7h). Our data support the conclusion that the phenotypic differences between WT and

FAM73b KO macrophages were correlated with metabolic reprogramming mediated by mitochondria.

Recent evidence has revealed that the balance between the utilization of glycolysis and mitochondrial respiration is controlled by NF-κB, which is also crucial for *Il12a* induction[43]. To elucidate the role of NF-κB in FAM73b KO-mediated *Il12a* augmentation, we examined MAP kinase (MAPK) and canonical NF-κB activation. Immunoblot detection indicated that the increase in *Il12a* is not due to hyperactivation of IKK, based on the level of phosphorylated IKKs, or its substrates IκBα and p105 (Fig. 5a). *Fam73b* deficiency also did not appreciably affect activation of the three major MAPK families (Fig. 5b). Furthermore, loss of FAM73b was dispensable for nuclear translocation of NF-κB members (c-Rel, p50, and p65), as revealed by the similar levels of these proteins in the nucleus (Fig. 5c). Our results suggested that *Il12a* augmentation was not due to hyperactivation of these conventional signaling pathways.

Previous studies revealed that the transcription factor IRF1 is required for IL-12 p35 induction and for suppressing IL-10 and IL-23 production, which is consistent with the phenotypes caused by FAM73b deficiency[44, 45]. Interestingly, immunoblot analyses revealed that both FAM73b (Fig. 5d) and FAM73a KO (Supplementary Fig. 8a) macrophages displayed a higher IRF1 protein level. Comparable levels of *Irf1* mRNA indicated that the IRF1 increase was associated with posttranslational modification (Fig. 5e). A proteasome inhibitor, MG132, largely rescued the IRF1 level in WT macrophages, suggesting that mitochondrial fission stabilized IRF1 by suppressing its proteolysis process (Fig. 5f). Consistently, loss of IL-12 mRNA expression was

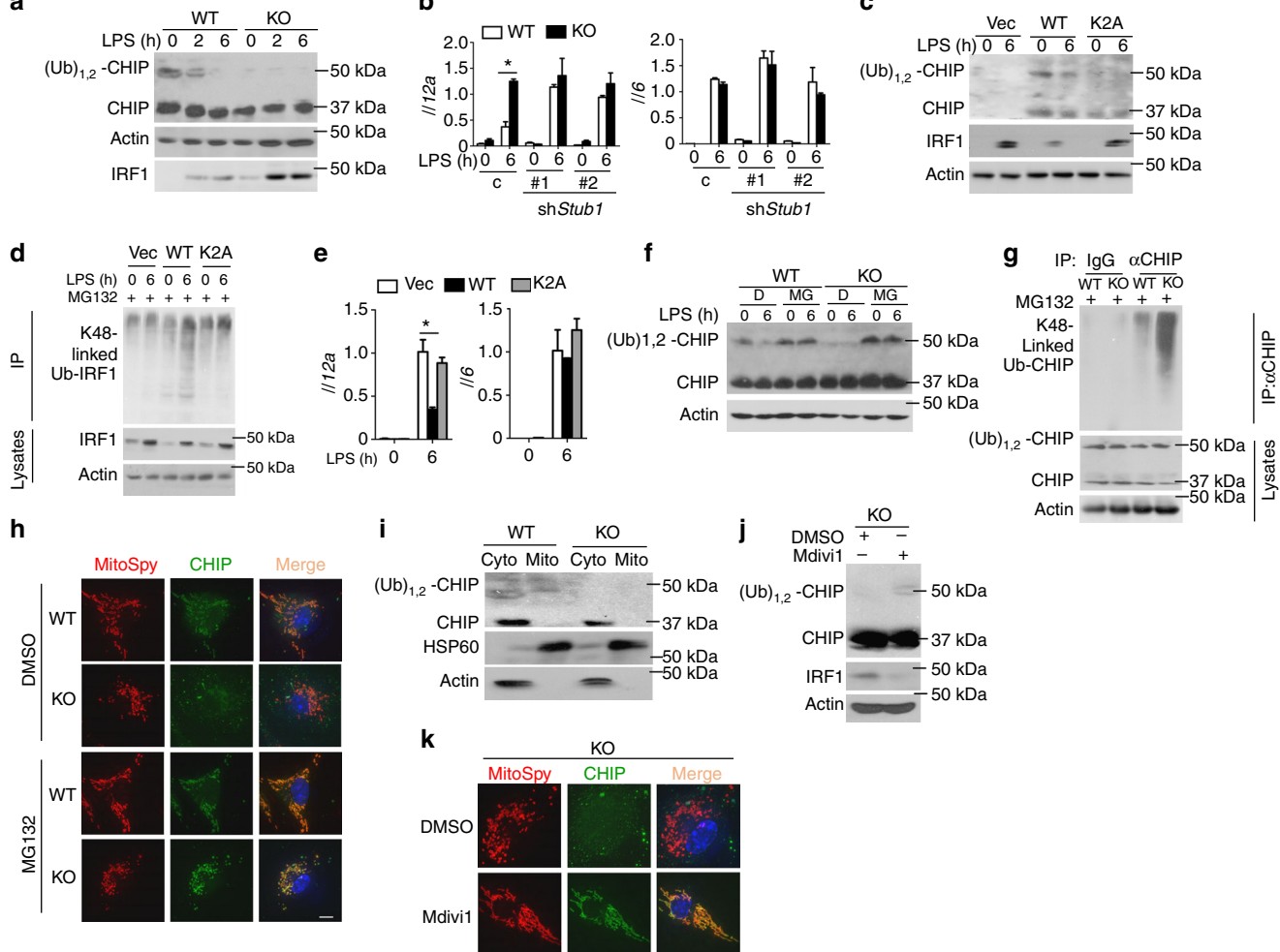

**Fig. 6** CHIP monoubiquitination is involved in IRF1 degradation. **a** IB analysis of CHIP and monoubiquitinated CHIP in the cytoplasm and IRF1 in the nucleus after stimulation with LPS as indicated. **b** WT or FAM73b KO BMDMs were infected with shRNA (C) or two different *Stub1*-specific shRNAs. qRT-PCR analysis of the indicated cytokines. **c**, **d** CHIP-deficient RAW264.7 cells were reconstituted with WT and mutant CHIP. **c** IB analyses of the indicated proteins in subcellular fractions. **d** IRF1 polyubiquitination in whole-cell lysates. **e** qRT-PCR analysis of *Il12a* and *Il6* mRNA in reconstituted RAW264.7 cells. **f**, **g** WT and FAM73b KO BMDMs were treated with MG132 for 30 min before being collected. IB analyses of CHIP monoubiquitination in the cytoplasm (**f**) and the polyubiquitination level (**g**). **h** Confocal microscopy analyses of CHIP location. **i** Immunoblot analysis of CHIP in mitochondrial or non-mitochondrial extracts. The purity of the mitochondrial fraction was confirmed using antibodies targeting HSP60 (a mitochondria marker) and Actin. **j**, **k** FAM73b KO BMDMs were treated with Mdivi-1 for 12 h. The protein level and location of CHIP were measured by IB (**j**) and confocal microscopy (**k**). The data are representative of three independent experiments. Scale bars in **h** and **k** are 10 μm. Statistical analyses represent variations in experimental replicates. Two-tailed unpaired *t*-tests were performed; *$P < 0.05$

observed in activated macrophages from IRF-1 KO mice (Fig. 5g). Further, to test the role of IRF1 aggregation, we used two different shRNAs to knockdown the *Irf1* gene in both WT and FAM73b KO BMDMs (Supplementary Fig. 8b). IRF1 silencing specifically impaired *Il12a* induction and erased the difference between WT and KO cells (Fig. 5h). Our results suggest that IRF1 plays an essential role in mitochondrial dynamics-mediated inflammatory responses. FAM73b KO BMDMs also had profoundly less abundant IRF1 K48-linked ubiquitination (Fig. 5i), suggesting a requirement of mitochondrial fission in suppressing IRF1 ubiquitination.

**Mitochondrial fission impairs CHIP monoubiquitination.** An E3 STUB1, known as CHIP, has been reported to mediate IRF1 protein degradation via ubiquitination[32]. A recent study established that a fraction of the CHIP population is

monoubiquitinated in vivo, which has functional consequences for CHIP activity[46]. Therefore, we compared the CHIP mono-ubiquitination levels between WT and FAM73b KO BMDMs. Surprisingly, monoubiquitinated CHIP was severely abolished in the cytoplasm of FAM73b KO BMDMs (Fig. 6a). CHIP knockdown elevated IRF1 accumulation and *Il12a* production in WT cells, but little or no effect was observed without *Fam73b* (Fig. 6b and Supplementary Fig. 8c). A monoubiquitination site, K2, of CHIP that resides in an N-terminal extension is conserved in mammals (Supplementary Fig. 8d). To investigate the function of this site, we knocked out endogenous CHIP in the murine macrophage cell line RAW264.7 using a CRISPR/CAS9 system. CHIP KO RAW264.7 cells were reconstituted with WT and mutant CHIP K2A. Mutant CHIP lacked monoubiquitination, which impaired its capacity to promote IRF1 K48-linked poly-ubiquitination (Fig. 6c, d). Additionally, mutant CHIP did not cripple *Il12a* induction as was seen in the WT control (Fig. 6e).

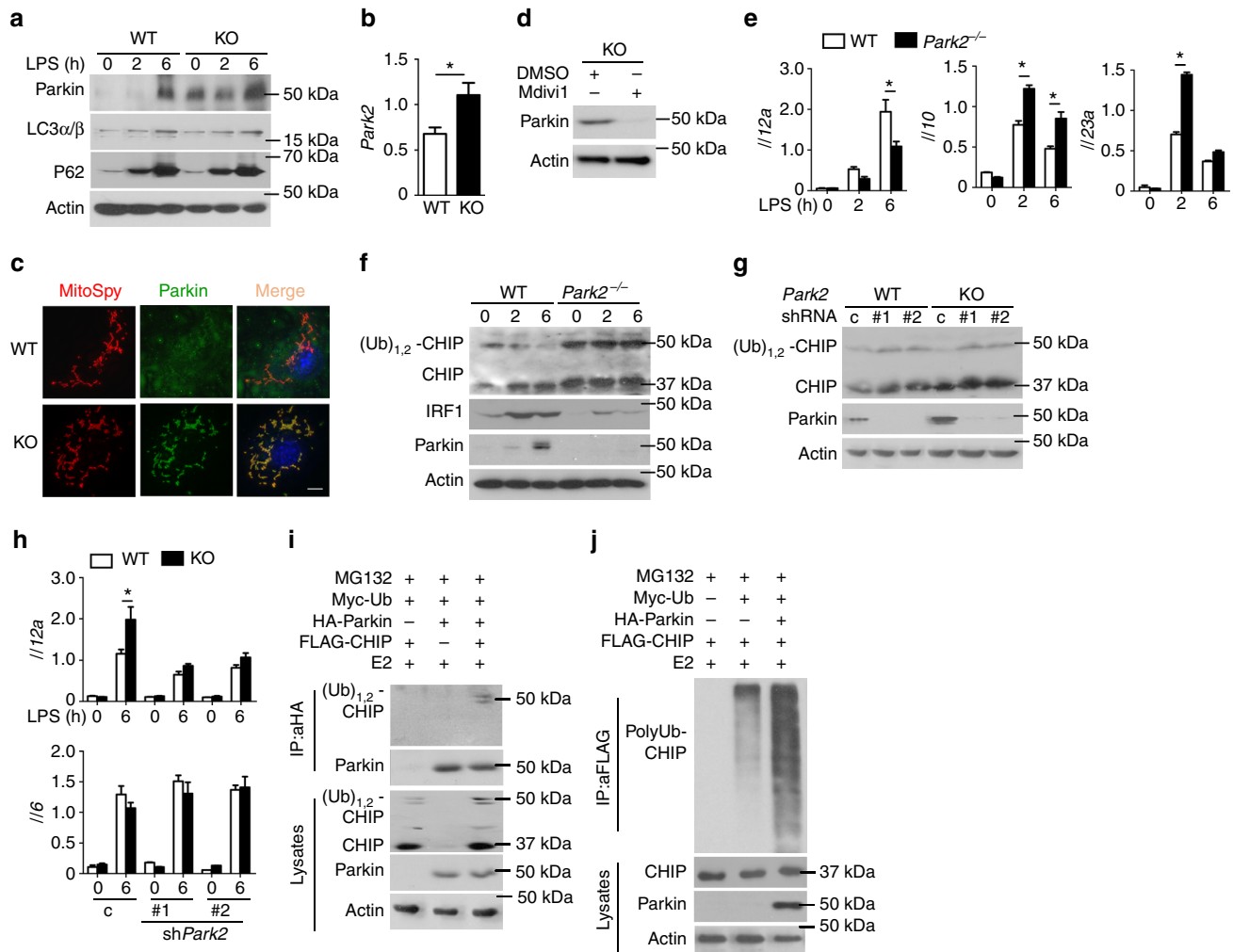

**Fig. 7** Parkin promotes degradation of monoubiquitinated CHIP. **a** IB analyses of Parkin and autophagy-related proteins in LPS-primed WT or FAM73b KO BMDMs. **b** *Park2* mRNA was detected by RT-QPCR. **c** Confocal microscopy analyses of Parkin and mitochondria colocalization. **d** FAM73b KO BMDMs were pretreated with Mdivi-1 for 12 h. The Parkin level was detected by IB. **e** Cytokine induction in Parkin KO BMDMs was measured by RT-QPCR. **f** IB analysis of CHIP in the cytoplasm and IRF1 protein level and modification. **g, h** Monoubiquitination of CHIP in the cytoplasm (**g**) and cytokine induction (**h**) were performed in *park2*-silenced BMDMs. **i** Parkin–CHIP interaction assays in HEK293T cells co-transfected with plasmids encoding the indicated genes. After treatment with MG132, whole-cell lysates were subjected to IP using anti-HA, followed by IB analysis of associated FLAG-CHIP using anti-FLAG. **j** HEK293T cells were transfected with total ubiquitin plasmid along with the indicated expression plasmids. FLAG-tagged CHIP was isolated by IP, followed by detection of polyubiquitinated CHIP by IB. The data are representative of three independent experiments. Scale bar in **c** is 10 μm. Two-tailed unpaired *t*-tests were performed; *P < 0.05

MG132 treatment indicated that degradation of mono-ubiquitinated CHIP mediated by mitochondrial fission mainly occurred through the proteolysis pathway (Fig. 6f). Furthermore, FAM73b KO BMDMs had profoundly more abundant CHIP K48-linked polyubiquitination when proteasome activity was blocked (Fig. 6g).

To examine the interaction between mitochondria and CHIP, we performed immunofluorescence image analysis using confocal microscopy. The results showed that CHIP did not locate at mitochondria in KO cells. This dissociation was rescued by MG132 treatment (Fig. 6h). Interestingly, subcellular fractionation studies also revealed that only monoubiquitinated CHIP was associated with mitochondria (Fig. 6i). Furthermore, Mdivi-1 treatment restored the monoubiquitinated CHIP level (Fig. 6j). CHIP recruitment to the mitochondria was also rescued by Mdivi-1 (Fig. 6k) in FAM73b KO BMDMs. Together, these results establish that fusion mitochondria-associated

monoubiquitinated CHIP is indispensable for IRF1 K48-linked polyubiquitination and degradation via proteolysis.

**Fission mitochondria degrade monoubiquitinated CHIP.** CHIP is known to be associated with Parkin, which is required for clearing damaged mitochondria to maintain cellular homeostasis[47]. Surprisingly, *Fam73b*- and *Mfn1/2*-deleted cells had high levels of Parkin (Fig. 7a and Supplementary Fig. 8e), suggesting that mitochondrial fission triggered increased Parkin expression. However, comparable p62 and light chain 3 (LC3α/β) levels indicated that CHIP degradation was not due to mitophagy activation (Fig. 7a). The increase in *Park2* mRNA abundance further indicated that mitochondrial fission enhanced the Parkin transcriptional level (Fig. 7b). Upregulated Parkin is selectively recruited to fission mitochondria in FAM73b-deficient cells (Fig. 7c). Interestingly, Mdivi-1 not only restored CHIP mono-ubiquitination but also suppressed Parkin expression (Fig. 7d). To

evaluate the role of Parkin in IL-12 induction, we collected Parkin KO BMDMs and measured various cytokine levels. Parkin deficiency suppressed IL-12 expression but promoted IL-10 and IL-23 induction (Fig. 7e). Loss of Parkin also upregulated CHIP monoubiquitination and inhibited IRF1 expression (Fig. 7f). Moreover, Parkin silencing significantly rescued the stability of monoubiquitinated CHIP and erased the phenotype driven by *Fam73b* depletion (Fig. 7g, h). Together, all of these data suggest that Parkin accumulation contributes to FAM73b KO-mediated phenotypes.

To clarify the relationship between Parkin and CHIP, we first evaluated their interaction. The results indicated that over-expressed Parkin mainly associated with monoubiquitinated CHIP but not regular CHIP in HEK293T cells (Fig. 7i). Furthermore, coexpression of CHIP with Parkin strongly induced CHIP polyubiquitination in HEK 293T cells (Fig. 7j). Taken together, these data suggest that mitochondrial fission upregulated the expression and recruitment of the crucial E3 ubiquitin ligase Parkin, which colocalized with CHIP and induced monoubiquitinated CHIP degradation.

## Discussion

Mitochondria are dynamic organelles that frequently divide and fuse[2, 48]. Previous evidence has revealed that mitochondria not only sustain immune cell homeostasis but are also necessary for launching immune responses. However, how mitochondrial dynamics determine the subtypes of immune responses remains poorly investigated. In this study, we clarified an essential role of the mitochondrial morphology in promoting IL-12 induction in innate immunity. The outer mitochondrial membrane protein FAM73b promotes mitochondrial fusion and suppresses TLR-stimulated IL-12 expression. Elevation of the FAM73b level is significantly associated with various types of stimulation. However, the classical fusion regulators *Mfn1*, *Mfn2*, and *Opa1* also induce phenotypes similar to *Fam73b* in macrophages. These results indicate that multiple mitochondrial dynamics-related proteins may be involved in innate immunity regulation. Interestingly, our data showed that *Fam73b* deficiency functioned as a negative regulator of TLR-induced type I IFNs, which was different from MFN1/2 KO cells[49]. Therefore, type I IFN regulation by *Mfn1/Mfn2* depends on their individual functions but not the morphology switch directly.

Current research suggests that mitochondrial dynamics control the balance between metabolic pathway engagement and T-cell fate[23]. However, as shown in Fig. 3a–d, *Fam73b* depletion did not have an intrinsic role in T-cell activation. In contrast to the *Opa1*−/− T cells, CD4+ or CD8+ T cells from T-cell-specific KO mice did not exhibit any increases in memory T cells or differences in CD4+ T-cell differentiation. These data indicate that the potential functions of mitochondrial dynamics in T cells remain incompletely investigated. These differences indicated that the phenotypes observed in *Opa1*−/− mice might be due to its individual functions but not the mitochondrial morphology.

Mitochondrial dynamics influence cellular function through multiple mechanisms, such as cell survival and metabolic output. Mitochondrial fusion is known to be important for efficient FAO via lipids, but fission promotes aerobic glycolysis[50, 51]. Furthermore, according to a previous finding, M2 macrophages exhibit enhanced FAO, while M1 macrophages rely instead on aerobic glycolysis[27]. However, few studies have elucidated the relationship between mitochondrial dynamics, metabolic changes and macrophage polarization. Our study found that *Fam73b* deficiency promoted M1-like phenotypes without an increase in ECAR. This result suggests a novel mechanism of mitochondrial morphology regulating macrophage polarization. Additionally,

molecular evidence revealed that mitochondrial fission facilitated IL-12 induction through regulation of the CHIP–IRF1 axis stability via Parkin. The Pink1–Parkin pathway has been demonstrated to promote mitochondrial fission by downregulating MFN1 and OPA1[52]. However, MFN1/2 or FAM73b deficiency also enhanced Parkin levels. These results suggest that mitochondrial fission is part of a feedback loop for promoting the Parkin level and maintaining mitochondria quality. ER stress has been shown to be important for activation of the unfolded protein response (UPR) and thereby for upregulation of Parkin expression[53]. FAM73b deficiency may enhance Parkin transcription by inducing ER stress. Based on our findings, we propose a model explaining how mitochondrial fission regulates TLR-stimulated *Il12a* expression (Supplementary Fig. 9).

In conclusion, our study presents a new concept of cancer immunotherapy through modulation of the mitochondrial morphology. Physically, TLRs and the suppressive cytokine IL-4 promotes mitochondrial fission and fusion, respectively, by regulating the *Fam73b* level. Mitochondrial fission is mediated by various molecules and specifically enhances *Il12a* induction and inhibits suppressive regulators such as *Il10* and *Arg1*. Both germline and myeloid cell-specific *Fam73b* ablation profoundly enhances T-cell responses to tumor growth. Our data further suggest a central function of mitochondrial fission in regulating IRF1 stability and promoting monoubiquitinated CHIP degradation. Moreover, upregulated Parkin recruits to fission mitochondria and inhibits CHIP–IRF1 axis signal transduction, which is in turn important for enhancing IFN-γ induction for Th1 polarization in vivo. Our findings demonstrate a novel signaling network that controls the innate immunity response and has profound therapeutic implications for cancer treatment.

## Methods

**Mice**. Mice were maintained under specific-pathogen-free conditions in a controlled environment of 20 °C–22 °C, with a 12/12 h light/dark cycle and 50–70% humidity; food and water were provided ad libitum. *Fam73a*-targeted mice were produced by transcription activator-like effector nucleases (TALEN) from the FVB/N strain. Parkin KO mice (Strain ID: 006582, C57BL/6 background) were provided by Dr. Shuo Yang from the Jackson Laboratory. IRF1 KO mice (Strain ID: 002762, B6.129S2 background) were purchased from the Jackson Laboratory. The heterozygous FAM73b-knockout first mice (Strain ID: Fam73b^tm1a(KOMP)Wtsi) were originally obtained from the UC DAVIS Knockout Mouse Project (KOMP) Repository. Male and female heterozygous FAM73b-knockout first mice (*Fam73b*+/−) were mated to each other to produce control WT and homozygous KO mice (*Fam73b*−/−). In FAM73b KO first mice, exon 3 of the *Fam73b* gene was targeted using a FRT-LoxP vector (Supplementary Fig. 2a). *Fam73b*-floxed mice were generated by crossing the FAM73b KO first mice with FLP deleter mice (Rosa26-FLPe; Jackson Laboratory). The *Fam73b*-floxed mice were further crossed with *Cd4*-Cre and *Lyz2*-Cre mice (all from Jackson Laboratory, C57BL/6 background) to generate T cell conditional FAM73b KO (*Fam73b*^f/f*Cd4*-Cre), myeloid cell conditional FAM73b KO (*Fam73b*^f/f*Lyz2*-Cre), and dendritic cell conditional FAM73b KO (*Fam73b*^f/f*Itgax*-Cre) mice. Heterozygous mice were bred to generate littermate controls and KO (or conditional KO) mice for experiments. In the animal studies, the WT and multiple KO mice are randomly grouped. Outcomes of animal experiments were collected blindly and recorded based on ear-tag numbers of 6- to 8-week-old experimental mice. Genotyping was performed as indicated in Supplementary Fig. 2b, f, g. All animal experiments were conducted in accordance with protocols approved by the Institutional Animal Care and Use Committee of Zhejiang University.

**Antibodies and reagents**. Antibodies targeting IKBα (C-21, 1:1000), Fam73b (S-12, 1:1000), p65 (C-20, 1:1000), Lamin B (C-20, 1:1000), ERK (K-23, 1:2000), phospho-ERK (E-4, 1:1000), JNK2 (C-17, 1:1000), p38 (H-147, 1:1000), IKKα (H-744, 1:1000), ubiquitin (P4D1, 1:1000), p105/p50 (C-19, 1:1000), IRF1 (M-20, 1:1000), CHIP (H-231, 1:1000), HSP60 (H-1, 1:3000), TBK1 (108A429, 1:1000), IRF3 (SC-9082, 1:1000), LC3α/β (SC-292354, 1:1000), and c-Rel (sc-71, 1:1000), as well as a control rabbit IgG (sc-2027), were from Santa Cruz Biotechnology. Antibodies targeting phospho-DRP1 (Ser637, #3455, 1:1000), DRP1 (D6C7, 1:1000), Mitofusin-2 (D1E9, 1:1000), phospho-IκBα (Ser32, 14D4, 1:1000), phospho-JNK (Thr180/Tyr185, #9251, 1:1000), phospho-p38 (Thr180/Tyr182, 3D7, 1:1000), phospho-p105 (Ser933, 18E6, 1:1000), phospho-IKKα/β (Ser176/180, 16A6, 1:1000), phospho-TBK1 (Ser172, D52C2, 1:1000), phospho-IRF3 (Ser396, 4D4G, 1:1000), and K48-linkage-specific polyubiquitin (D9D5, 1:8000) were

purchased from Cell Signaling Technology Inc. Anti-actin (C-4, 1:10,000) was from Sigma. HRP-conjugated anti-HA (HA-7) and anti-FLAG (M2) were from Sigma-Aldrich. Antibodies targeting p62 (ab56416, 1:1000) and Parkin (ab15954, 1:1000) were from Abcam. Fluorescence-labeled antibodies are listed in the section describing the flow cytometry and cell sorting procedures.

Mouse CHIP and Parkin cDNA were amplified from splenic mouse mRNA using PCR and inserted into the pCLXSN(GFP) retroviral vector. HA-Ub, Myc-Ub, and subtypes of Ub expression plasmids were provided by Dr. Shao-cong Sun. pGIPZ lentiviral vectors encoding a non-silencing shRNA control and two different *Irf1*, *Stub1*, and *Park2* shRNAs were designed and produced by Invitrogen.

LPS (derived from *Escherichia coli* strain 0127:B8) and CpG (2216) were from Sigma-Aldrich. R848 and Poly I:C was from Amersham, and recombinant murine M-CSF was from Peprotech.

All the uncropped scans of the western blots are shown in Supplementary Fig. 10.

**Flow cytometry and ICS**. Single-cell suspensions from B16 melanoma, spleens or draining lymph nodes were subjected to flow cytometry using CytoFlex (Beckman Coulter) and the following fluorescence-labeled antibodies from eBioscience: PB-conjugated anti-CD4 and anti-CD11c; PE-conjugated anti-B220 and anti-F4/80; PerCP5.5-conjugated anti-Gr-1 (Ly6G); APC-conjugated anti-CD62L; APC-CY7-conjugated anti-CD11b and anti-CD8; and FITC-conjugated anti-IFNγ, anti-CD44 and anti-Foxp3. DAPI was from Life Technologies, and MitoSpy$^{TM}$ Orange CMTMRos was from BioLegend.

For intracellular cytokine staining (ICS), the tumor-infiltrating T cells were stimulated for 4 h with PMA plus ionomycin in the presence of monensin and then subjected to intracellular IFN-γ and subsequent flow cytometry analysis. All FACS data were analyzed by FlowJo 7.6.1. All FACS gating/sorting strategies are shown in Supplementary Fig. 11.

**Fluorescence microscopy**. BMDMs ($5 \times 10^5$) were collected and seeded on 12-well plates containing 70% alcohol-pretreated slides for starvation overnight. Cells treated with or without stimulation as indicated were stained with 250 nM MitoSpy$^{TM}$ Orange CMTMRos for 20 min and fixed with 4% paraformaldehyde (PFA) for 20 min. Then, the cells were washed with PBS three times and stained with 10 µg/ml DAPI. All the samples were imaged on a LSM710 (Carl Zeiss) confocal microscope outfitted with a Plan-Apochromat ×63 oil immersion objective lens (Carl Zeiss). The data were collected using Carl Zeiss software ZEN 2010. To quantify the mitochondrial morphology of MitoSpy$^{TM}$ Orange CMTMRos-stained macrophages, scoring was blindly analyzed with Image-Pro software. Short was defined as cells with a majority of mitochondria less than 7 µm and long as cells in which the majority of mitochondria were more than 7 µm.

**Transmission electron microscopy**. WT and *Fam73b* KO macrophages were washed in PBS and fixed in 2.5% GA on ice for 15 min. Then, the cells were scraped and put into a 1.5 ml EP tube. The GA solution was refreshed, and the cells were suspended and incubated at 4 °C overnight. Then, the cells were embedded into agarose gel. The gel was cut into small pieces, washed in PBS and post-fixed in 1% osmic acid for 1–2 h. Then, the samples were washed in PBS and dehydrated in a gradient ethanol series (50%, 75%, 85%, 95% and 100% ethanol), each for 15 min. Then, the samples were embedded in Epon resin. Embedded samples were cut into 60-nm ultrathin sections, and sections were counterstained with uranyl acetate and lead citrate. All the samples were observed using a Hitachi HT7700 electron microscope.

**Time-lapse microscopy**. BMDMs ($2 \times 10^5$) were plated in glass chamber slides (LabTek) and starved overnight. Cells were stained for 20 min with 250 nM MitoSpy$^{TM}$ Orange CMTMRos (BioLegend). Imaging was started immediately after addition of 500 ng/ml LPS on a DeltaVision RT system with SoftWorx software (Applied Precision) at 37 °C and 10% $CO_2$. Images were acquired every 10 min for 2 h. The data were exported as uncompressed AVI files and processed with Premiere Pro.

**B16 model of melanoma**. WT, FAM73b KO, *Fam73b*-*lyz2*$^{cre/+}$, *Fam73b*-*Itgax*$^{cre/+}$and *Fam73b*-*cd4*$^{cre/+}$ mice administered a s.c. injection of $2 \times 10^5$ B16 melanoma cells. Tumor size is presented as a square caliper measurement calculated based on two perpendicular diameters (mm$^2$). The injected mice were monitored for tumor growth every other day, and based on protocols approved by the IACUC of Zhejiang University. The maximum sizes of tumors are limited up to 225 mm$^2$ by ethical permission. Thus, the mice were sacrificed and defined as lethality when the tumor size reached to 225 mm$^2$. For analysis of tumor growth rate and tumor-infiltrating immune cells, a lower number ($2 \times 10^5$) of B16 cells were injected to prevent lethality during the course of the experiment. To minimize individual variations, littermate WT and different *Fam73b* conditional KO mice were used. Mice were randomly selected for tumor injection, and analysis of tumor size was performed in a blinded fashion. To analyze tumor-infiltrating immune cells, tumor tissues were treated with 0.25 mg/ml collagenase A (Sigma-Aldrich) and 25 U/ml DNase (Roche Diagnostics, Indianapolis, IN) for 20 min at 37 °C, and the cells were passed through a plastic mesh. The resulting dissociated cells were collected by centrifugation, resuspended in Red Blood Cell Lysis Buffer for 3 min, and washed twice in PBS. The cells were subjected to flow cytometry analysis, and TAMs and MDSCs were gated based on CD11b$^+$F4/80$^+$ or CD11b$^+$Gr-1$^{high}$ surface markers, respectively.

**MCA-induced fibrosarcoma**. WT and *Fam73b* KO mice were given a s.c. injection of a high dose of MCA (800 µg), as indicated in the figure legends, in the right flank. The mice were monitored for fibrosarcoma development weekly over the course of 150 days. Tumor size is presented as a square caliper measurement calculated based on two perpendicular diameters (mm$^2$). The mice were monitored for tumor growth every week. Mice with tumors larger than 225 mm$^2$ were sacrificed and recorded as having lethal tumors based on protocols approved by the Institutional Animal Care and Use Committee of Zhejiang University.

**Generation of BMDCs and BMDMs**. Bone marrow cells isolated from the WT or dendritic cell-specific KO mice were cultured in RPMI 1640 medium containing 10% FBS supplemented with GM-CSF (10 ng/ml) for 7 days. The differentiated DCs were stained with Pacific blue-conjugated anti-CD11c and isolated with a FACS sorter. In some experiments, BMDMs were generated using MCSF-supplemented medium.

**ROS assay**. WT and *Fam73b* KO macrophages were washed with PBS twice and stained with DCFH-DA (10 µM; Beyotime) in DMEM without FBS for 20 min at 37 °C. After being washed with PBS twice, the cells were harvested and analyzed by flow cytometry (Cytoplex, Beckman Coulter).

**Metabolism assays**. The OCR and ECAR were measured in XFp extracellular flux analyzers (EFAs) (Seahorse Bioscience) using a XFp Cell Mito Stress Test Kit and a XFp Glycolysis Stress Test kit, respectively. The following parameters were used in the assays: seed cells $8 \times 10^4$ per well, Oligomycin 1.0 µM, FCCP 1.0 µM, Rotenone/antimycin A 0.5 µM, glucose 10 mM, and 2-DG 50 mM as indicated.

**shRNA knockdown**. Lentiviral particles were produced by transfecting HEK293T cells (using the calcium-phosphate method) with a pGIPZ lentiviral vector encoding either a non-silencing shRNA or *Irf1*-, *Stub1*-, or *Park2*-specific shRNA, along with the packaging vectors psPAX2 and pMD2. BMDMs, differentiated using M-CSF-supplemented medium for 5 days, were infected with the lentiviruses for 8 h. After 72 h, the infected cells were enriched via flow cytometric cell sorting (based on GFP expression) and subsequently used for experiments.

**ELISA and qRT-PCR**. Supernatants of in vitro cell cultures were analyzed via ELISA using a commercial assay system (eBioScience). For qRT-PCR, total RNA was isolated using TRIzol reagent (Molecular Research Center, Inc.) and subjected to cDNA synthesis using RNase H-reverse transcriptase (Invitrogen) and oligo (dT) primers. qRT-PCR was performed in triplicate using an iCycler Sequence Detection System (Bio-Rad) and iQTM SYBR Green Supermix (Bio-Rad). The expression of individual genes was calculated with a standard curve and normalized to the expression of *Actb*. The gene-specific PCR primers (all for mouse genes) are shown in Supplementary Table 1.

**Ubiquitination assays**. Cells were pretreated with MG132 for 2 h and then lysed with a Nonidet P-40 lysis buffer (50 mM Tris-HCl, pH 7.5, 120 mM NaCl, 1% Nonidet P-40, 1 mM EDTA, and 1 mM DTT) containing 6 M urea and protease inhibitors. IRF1 or CHIP was isolated by IP with antibodies targeting IRF1 (M-20) and CHIP (H-231). The ubiquitinated proteins were detected by IB using an anti-ubiquitin (Santa, P4D1) or anti-K48-linked ubiquitin (Millipore, 05–1307) antibody.

**Statistical analysis**. Statistical analysis was performed using Prism software. No data are excluded from the analyses. Two-tailed unpaired $t$-tests were performed. $P$ values <0.05 were considered significant, and the level of significance is indicated as *$P < 0.05$ and **$P < 0.01$. In the animal studies, a minimum of four mice were required for each group based on the calculated number necessary to achieve a 2.3-fold change (effect size) in a two-tailed $t$-test with 90% power and a significance level of 5%. All statistical tests are justified as appropriate, and the data meet the assumptions of the tests. The variance is similar between the statistically compared groups.

**Data availability**. Sequence data that support the findings of this study are available from the authors and have been deposited in the National Center for Biotechnology Information (NCBI) BioProject with the primary accession code PRJNA359723. All other data are available from the authors upon reasonable requests.

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

## Acknowledgements

We thank Dr. Shao-cong Sun, Dr. Zongping Xia, Dr. Bin Zhao, and Dr. Hong-bing Shu for the expression vectors. This study was supported by the National Natural Science Foundation of China (81572651/81771675), Fundamental Research Funds for the Central Universities (2016QN81013), the National Natural Science Foundation of China (31471315/ 31671457), the Thousand Young Talents Plan of China, and the Zhejiang University Special Fund for Fundamental Research.

## Author contributions

Z.G. and Y.L. performed the research, prepared the figures, and wrote the manuscript; F. W., T.H., K.F., Y.Z. and J.Z. contributed to the experiments; Q.C., C.T., J.J, S.Y., L.Z., Y.

X., J.-Y.Z. and X.-H.F. contributed to the experimental design and wrote the manuscript; J.J. supervised the work, prepared the figures, and wrote the manuscript.

## Additional information

**Competing interests:** The authors declare no competing financial interests.

