## [Peer Review File · Nature Communications]

Reviewers' comments:

Reviewer #1 (Remarks to the Author):

In this manuscript, Gao et al begin with analysis of Fam73b in macrophages. LPS and IL-4 have different effects on Fam73b gene expression that correlate with their differential effects on mito fusion/fission dynamics. Consistent with a role for Fam73b in regulating mito fusion, Fam73b KO BMDMs have fragmented mito. Interestingly, KO BMDMs have increased LPS-inducible expression of Ii12a but reduced LPS-inducible expression of Ii10 and Ii23a. In tumor models, Fam73b deficiency in the myeloid compartment dramatically reduced tumor burden and increased mice survival. This is attributed to increased production of IL-12 in TAMs, which leads to increased IFN- γ production by CD4 and CD8 T cells in the tumor and in draining lymph nodes. Moreover, pharmacological treatment with Mdivi to enforce fusion ablates the differences in Ii12a and IL-10 induction between WT and KO macs, suggesting that fission per se critically underlies the phenotype of KO macs. These data are clear and striking and straightforward to interpret.

The manuscript next address the mechanism by which Fam73b deficiency leads to these phenotypes. Increased IRF1 levels in the KO BMDMs may be a key factor, since IRF1 is known to positively regulate IL-12 production. From here the story gets increasingly complicated and difficult to follow, and the quality of key pieces of data are not strong (see below). The proposed mechanism seems to involve regulation of IRF1 protein stability/degradation, via CHIP (previously implicated in IRF1 protein stability), Ube2w (previously implicated in regulation of CHIP monoubiquitination), and Parkin (previously shown to associate with CHIP).

It is not clear to me why the authors chose to go down this laborious route in the last third of the manuscript. They made a KO of Fam73b and have a very striking phenotype in vitro as well as in vivo. They implicate IRF1 and mito fission/fusion in the phenotype. By jumping to CHIP, Ube2w, and Parkin, the manuscript rapidly loses its focus on Fam73b (for which the data is very nice), in an effort to establish connections between some proteins that are not well characterized previously or in this manuscript. For this reviewer, most of the data in the last two figures do not add to and instead detract from the story.

There is some interesting material here. Unfortunately, the story becomes mired in an increasingly complicated pathway that remains poorly characterized and for which some key pieces of data are not very strong or well-explained. I suggest that the reviewers trim some of the data in the last third of the manuscript to keep the story focused on the role of Fam73b and mitochondrial fission/fusion in regulation of IRF1 levels and consequent effects on IL-12 production. In addition, it is recommended that the authors seek an editing service to correct typos and improve the English/grammar and readability of the manuscript.

Specific comments:

1) Some of the western blots are not very convincing:

- Fig 5h: It looks like there is less, not more, IRF1 in the KO macs.
- In Fig 5d, it looks like the big difference between WT and KO macs is in LPS-inducible levels of IRF1, but in Fig 6a, the difference between WT and KO is in steady state levels of IRF1.
- Most of the ChIP-Ub blots are not entirely convincing.
- I do not see evidence for LPS-induced degradation of CHIP in their western blots.

2) The authors need to better introduce Fam73b in the Introduction section, since this is a major focus of the story and most readers would not be familiar with the protein. Likewise, if the author decides to keep CHIP and Ube2w in the story, they need to be better introduced so that the reader understands the biological relevance of their control of IL12a gene induction.

3) Line 346: "IRF1 silencing specifically promoted Ii12a induction" is the opposite of what the authors want to say.

4) Line 338: wrong call out of Fig S7b

5) Fig S6f: What is the difference between the left and right panels?

Reviewer #2 (Remarks to the Author):

In this manuscript, the authors report that mitochondrial fission, in an ubiquitin-dependent manner, stimulates IL-12 expression while inhibits that of IL-10 and IL-23 without affecting other proinflammatory cytokines. They observed that stimulation of BMDM with LPS or poly I:C promotes a rapid mitochondrial fission by downregulating the mitochondrial fusion effectors Mfn1/2 and the downstream effector Fam73b (initially named mitogardin or Miga). Genetic ablation of Fam73b confirmed that this mitochondrial protein is required in TLRs-regulated fragmentation/fission of the mitochondrial network and this phenotype specifically promotes IL-12 production. At the molecular level, the authors have observed that mitochondrial fission triggers the accumulation and recruitment of the E3 ubiquitin ligase Parkin at the mitochondria. Parkin then induces the degradation of monoubiquitinated CHIP and stabilizes the transcription factor IRF1 that is involved in IL-12 production. In vivo, this increased IL-12 production by macrophages enhances the antitumor T cells responses in a murine melanoma model as well as in a MCA-induced fibrosarcoma model. These findings suggest therefore an unexpected role of mitochondrial dynamics in cytokine production and ensuing anti-tumoral immunity. This study is really extensive and complete while sometime, quite complicated. Moreover I have few concerns.

To decipher the molecular mechanisms promoting IL-12 production downstream of mitochondrial fission, the authors have knocked down IRF1, Parkin and CHIP using shRNA in WT or Fam73 KO BMDM. As KO mice are available for IRF1 and Parkin, I guess that their conclusions should be confirmed by using BMDM from these KO mice. I mean for instance, IL12 production in IRF1 KO BMDM should be less important than in WT BMDM after TLR4 stimulation. The use of Parkin KO BMDM would be also interesting.

Specific comments:

In Fig 1a. It appears that 50 % of the BMDM have already short/fragmented mitochondria. TLR4 stimulation for 2 hrs with 500 ng/ml of LPS leads to a 20% increase in mitochondrial fragmentation. It does not seem to be the extensive mitochondrial fragmentation described in Fig 1c with time-lapse microscopy. 500 ng/ml of LPS is also quite a high concentration to stimulate BMDM. Do lower concentrations of LPS also trigger mitochondrial fragmentation? In fig 1g, the authors have assessed mitochondrial morphology 12 hrs after IL-4 stimulation. The mitochondrial network then appears fused. How is the mitochondrial network 12 hrs after TLR4 or TLR3 stimulation?

The data shown in Fig5h are not convincing. Indeed, I do not really see less K48-linked ubiquitinated IRF1 in Fam73b KO BMDMs.

In Fig 7k, it is not that obvious that Parkin preferentially promotes K48-linked ubiquitination of CHIP.

Does Mdivi1 affects Parkin expression and CHIP monoubiquitination/mitochondrial recruitment in Fam73b KO cells?

Minor comments:

Fig 2c is not mentioned in the text.

There are some typos and mistakes in the text. They should be corrected. Moreover, the English should be also edited.

The authors have observed an increase of Parkin in Fam73 or Mfns KO cells. Do the authors have a hypothesis to explain it?

This sentence "IRF1 silencing specifically promoted IL12a induction, and erased the differences

between WT and KO cells (Fig. 5g) » (p16, line 346) does not seem to be in agreement with the data shown actually in Fig 5g. Indeed, IRF1 silencing does not promote IL12a induction but I agree, erases the difference between WT and KO cells.
Line 564, the reference is not complete.

Reviewer #1:

Main concerns:

1. *It is not clear to me why the authors chose to go down this laborious route in the last third of the manuscript. They made a KO of Fam73b and have a very striking phenotype in vitro as well as in vivo. They implicate IRF1 and mito fission/fusion in the phenotype. By jumping to CHIP, Ube2w, and Parkin, the manuscript rapidly loses its focus on Fam73b (for which the data is very nice), in an effort to establish connections between some proteins that are not well characterized previously or in this manuscript. For this reviewer, most of the data in the last two figures do not add to and instead detract from the story.*

Response:

We thank the reviewer for the constructive comments and suggestions. Following the reviewer's suggestion, we delete some parts of complicated data in last two figures such as all Ube2w data and polyubiquitinated sites by Parkin, which are not associated with the final conclusion tightly. Furthermore, we apologize for not clearly describing the data linked first 4 Figures to last 3. We observe similar phenotypes among *Fam73b* KO (**Figure. 4c**), *Mfn1/2* DKO (**Figure. 4h**), *Opa1* KO (**Figure. 4i**) and *Fam73a* KO (**supplementary Figure. 6e**) cells, which indicates that IL-12a increment is due to the mitochondrial fission, but not special function mediated by *Fam73b*. In last two figures, we focus on the mechanism how mitochondrial fission regulates IRF1 accumulation. So we still remain some parts data of CHIP monoubiquitination and Parkin function, which well character IRF1 accumulation.

2. *There is some interesting material here. Unfortunately, the story becomes mired in an increasingly complicated pathway that remains poorly characterized and for which some key pieces of data are not very strong or well-explained. I suggest that*

the reviewers trim some of the data in the last third of the manuscript to keep the story focused on the role of Fam73b and mitochondrial fission/fusion in regulation of IRF1 levels and consequent effects on IL-12 production. In addition, it is recommended that the authors seek an editing service to correct typos and improve the English/grammar and readability of the manuscript.

Response:

Following this excellent suggestion, we have removed some data which is not associated with our crucial conclusion as mentioned above. Mitochondrial fission mediated by all of *Fam73a*, *Fam73b*, *Mfn1/2* *Opa1* deficiency performs the similar phenotypes, indicating mitochondrial morphology is required for IL-12 regulation, but not special function of one molecule. Therefore, we still keep the data of CHIP monoubiquitination and Parkin to clarify the underlying mechanism.

To improve the readability of our manuscript, we reorganize the data and correct typos with a language editing service.

Specif comment:

1. Some of the western blots are not very convincing:

- *Fig 5h: It looks like there is less, not more, IRF1 in the KO macs.*

Response:

We apologize for not clearly describing our data. In ubiquitination assay, we always pretreated with MG132 to block proteasome activity. MG132 treatment promotes IRF1 stability and normalizes the difference between WT and KO BMDMs. This data is consisting with IB shown in **Figure. 5f**.

• *In Fig 5d, it looks like the big difference between WT and KO macs is in LPS-inducible levels of IRF1, but in Fig 6a, the difference between WT and KO is in steady state levels of IRF1.*

Response:

The reviewer's point is well taken. We first perform the IRF1 level with longer exposure in **Fig.5d**. The data indicate steady state level of IRF1 in KO cells remains higher than WT one. Furthermore, we agree with the reviewer's concern for the difference of IRF1 in steady state in **Fig. 6a**. Actually, we have already repeated these results for other twice. We replace the IRF1 IB data with our repeating data. The new data together indicate both of basal and inducible IRF1 are upregulated in *Fam73b* KO BMDMs.

- *Most of the ChIP-Ub blots are not entirely convincing.*

Response:

We apologize for not presenting CHIP-Ub data with best quality. However, we are still confident for our conclusion, because the specific band of monoubiquitinated and regular CHIP disappeared when silencing its expression with shRNA in **supplementary Fig. 8c**. Furthermore, to respond to the reviewer's concern, we repeated some crucial ChIP-Ub blots as shown in **Fig. 6a, 6f**. In Parkin KO BMDMs, we also evaluated ChIP-Ub level (**Fig. 7f**). All of these results indicate that

monoubiquitinated CHIP degradation is regulated by Parkin and tightly control IRF1 stability.

- *I do not see evidence for LPS-induced degradation of CHIP in their western blots.*

Response:

This is a quite excellent question. Based on previous data in **Fig. 6a**, monoUbi-CHIP performs modern degradation in total cell lysates. However, due to the colocalization with mitochondria, we assume that monoUbi-CHIP may mainly be degraded in cytoplasm. Therefore, we carefully evaluated the level of monoUbi-CHIP in cytoplasm with LPS stimulation. As shown in supplementary **Fig.8d**, the results indicated that monoUbi-CHIP was degraded more significantly in cytoplasm than total cell lysates.

- *The authors need to better introduce Fam73b in the Introduction section, since this is a major focus of the story and most readers would not be familiar with the protein. Likewise, if the author decides to keep CHIP and Ube2w in the story, they need to be better introduced so that the reader understands the biological relevance of their control of IL12a gene induction.*

Response:

The reviewer's point is well taken. We add some introductions of *Fam73b* and CHIP biological function on IRF1 stability in first section on Page 3 and Page 4-5. In additionally, we follow the reviewer suggestions to delete all Ube2w data to make the story more clearly.

- *Line 346: "IRF1 silencing specifically promoted Il12a induction" is the opposite of what the authors want to say.*

Response:

We apologize for the error, and we have made the correction.

- *Line 338: Wrong call out of Fig S7b*

Response:

We apologize for not explaining the results clearly. Actually, we find both of *Fam73a* and *Fam73b* are potential suppressors for IRF1 stability. We replace the description as "Interestingly, IB analyses revealed that both of *Fam73b* (**Fig. 5d**) and *Fam73a* KO (**Supplementary Fig. 8a**) macrophages displayed a higher IRF1 protein level".

- *Fig S6f: What is the difference between the left and right panels?*

Response:

We apologize for not labeling the results clearly in **supplementary Figure 7e**. The left panel is basal level of OCR. The right one is OCR level with LPS stimulation for 6 hours. We add the legends back for these two panels.

Reviewer #2:

Main concerns:

To decipher the molecular mechanisms promoting IL-12 production downstream of mitochondrial fission, the authors have knocked down IRF1, Parkin and CHIP using shRNA in WT or Fam73 KO BMDM. As KO mice are available for IRF1 and Parkin, I guess that their conclusions should be confirmed by using BMDM from these KO mice. I mean for instance, IL12 production in IRF1 KO BMDM should be less important than in WT BMDM after TLR4 stimulation. The use of Parkin KO BMDM would be also interesting.

Response:

This is an excellent question. Following the reviewer's suggestion, we carefully evaluate cytokine induction either in IRF1 KO or Parkin KO BMDMs. Consistent with our conclusion, loss of IL-12 mRNA expression is observed in activated macrophages from IRF-1^{-/-} mice (**Fig. 5g**). Furthermore, Parkin deficiency suppresses IL-12 expression, but promotes IL-10 and IL-23 induction (**Fig. 7e**). Depletion of Parkin also stabilizes monoubiquitinated CHIP and inhibits IRF1 level (**Fig. 7f**). These data together suggest that Parkin-IRF1 signal pathway contributes to *Fam73b* KO-mediated sorts of phenotypes.

Specific comments:

• In Fig 1a. It appears that 50 % of the BMDM have already short/fragmented mitochondria. TLR4 stimulation for 2 hrs with 500ng/ml of LPS leads to a 20% increase in mitochondrial fragmentation. It does not seem to be the extensive mitochondrial fragmentation described in Fig 1c with time-lapse microscopy.

Response:

We apologize for not clearly describing the data. We do not evaluate the extension of mitochondrial fission with time-lapse microscopy. We have already modified the text in this section.

• 500 ng/ml of LPS is also quite a high concentration to stimulate BMDM. Do lower concentrations of LPS also trigger mitochondrial fragmentation?

Response:

This is an excellent question. Following the reviewer's suggestion, we carefully detect the effect of LPS with lower doses (10ng/ml and 100ng/ml). As shown in **supplementary Fig. 1a**, the intensity of mitochondrial fragment is dose dependent. The high dose LPS do not trigger mitochondrial fragmentation unlimited, because the dose with 100ng/ml have already performed similar phenotype of mitochondrial fission.

• In fig 1g, the authors have assessed mitochondrial morphology 12 hrs after IL-4 stimulation. The mitochondrial network then appears fused. How is the mitochondrial network 12 hrs after TLR4 or TLR3 stimulation?

Response:

As reviewer's suggestion, we perform the experiment and present the results in

supplementary Figure 1b. The results demonstrate mitochondrial network maintained fission status until 12hrs after stimulation. These data indicate the differences between WT and KO cells are not due to time extension under IL-4 and TLRs stimulation.

• *The data shown in Fig5h are not convincing. Indeed, I do not really see less K48-linked ubiquitinated IRF1 in Fam73b KO BMDMs.*

Response:

To respond to the reviewer's concern, we performed new experiments. The new data showed that IRF1 K48-linked ubiquitination was significantly impaired with Fam73b deficiency in **Fig. 5i**.

• *In Fig 7k, it is not that obvious that Parkin preferentially promotes K48-linked ubiquitination of CHIP.*

Response:

To respond to the reviewer's question, we perform new experiments and show that Parkin significantly promoted K48-linked ubiquitination of CHIP in **Fig. 7k**. However, we still observe a moderate enhancement of CHIP polyUbiquitination in K48R group. This data indicate Parkin also enhance other subtypes of CHIP polyUbiquitination. Therefore, we modified our conclusion that CHIP was preferred but not only to select K48-linked polyubiquitin chains by Parkin (**Fig. 7k**).

• *Does Mdivi1 affects Parkin expression and CHIP monoubiquitination/mitochondrial recruitment in Fam73b KO cells?*

Response:

To address the reviewer's question, we perform new experiments and show that Mdivi1 treatment significantly suppresses Parkin expression and stabilize CHIP monoubiquitination in *Fam73b* KO BMDMs (**Fig.6g**). In additionally, CHIP recruitment is also restored due to mitochondrial fusion (**Fig.6h**).

Minor comments:

• *Fig 2c is not mentioned in the text.*

Response:

We thank the reviewer for this remind, we have corrected this typo.

• *There are some typos and mistakes in the text. They should be corrected. Moreover, the English should be also edited.*

Response:

We thank the reviewer for this critical point. To improve the readability of our manuscript, we reorganize our data and correct typos with a language editing service.

• *The authors have observed an increase of Parkin in Fam73b or Mfns KO cells. Do the authors have a hypothesis to explain it?*

Response:

This is a quite interesting question. To respond to the reviewer's question, we first evaluate *Park2* mRNA level with RT-QPCR. Increment of *Park2* mRNA abundance indicates mitochondrial fission enhances Parkin expression in transcriptional level. As previous report, ER stress has been proved to be important for leading to the activation of the UPR, and thereby to an upregulation of Parkin expression¹. The same as MFNs, *Fam73b* also localizes at the area of mitochondria membrane contact with ER (data not shown). *Fam73b* deficiency may enhance Parkin transcription by inducing ER stress.

• *This sentence "IRF1 silencing specifically promoted Il12a induction, and erased the differences between WT and KO cells (Fig. 5g) » (p16, line 346) does not seem to be in agreement with the data shown actually in Fig 5g. Indeed, IRF1 silencing does not promote IL12a induction but I agree, erases the difference between WT and KO cells.*

Response:

We apologize for the error, and we have made the correction.

• *Line 564, the reference is not complete.*

Response:

We have corrected this typo.

Reference:

1. Bouman L, Schlierf A, Lutz AK, Shan J, Deinlein A, Kast J, *et al.* Parkin is transcriptionally regulated by ATF4: evidence for an interconnection between mitochondrial stress and ER stress. *Cell Death Differ* 2011, **18**(5): 769-782.

Reviewers' comments:

Reviewer #1 (Remarks to the Author):

The revised manuscript is improved, however some concerns remain:

- 1) The English remains a significant barrier to understanding the manuscript.
- 2) Fig 6i: I do not see association of monoubiquitinated CHIP with the mito.
- 3) Fig 7k: I do not understand what the authors mean by "We found that CHIP was preferred but not only to select K48-linked polyubiquitin chains by Parkin". The immunoblot seems to suggest no difference between HA-K48 and HA-K48R, but most importantly, I do not understand the rationale for this experiment. How would the authors interpret the data if there was a difference or if there was no difference between HA-K48 and HA-K48R? If there is no strong rationale for this experiment, the authors should consider not including the data in the manuscript.
- 4) Check lines 330-332 for figure call-outs.
- 5) Fig 1e: The authors should explain how they interpret LPS-inducible loss of Drp1 phosphorylation.

Reviewer #2 (Remarks to the Author):

The concerns raised by the referees have been addressed in a satisfactory manner. According to me, this manuscript is nearly suitable for publication. In line 268, the authors wrote that they have examined activation of MAPKs, NF-kB and STAT3. In figure 5, I do not see the analysis of STAT3 activation. I guess that in figure 7, panel "B" should be "C" and "C" should be "B" to be in the right order according to the text. Line 331, "(Fig. 7b)" is lacking. Moreover, again, some typos remain in the text and the English should be also edited.

In figure 2J, the values in the different histograms are not clear.

Line 344-345, the authors did use HEK293 or HEK293T?

Reviewer #1:

1. *The English remains a significant barrier to understanding the manuscript.*

Response:

To improve the readability of our manuscript, we used “Nature Publishing Group Language Editing” Golden service to make our manuscript clear and well written.

2. *Fig 6i: I do not see association of monoubiquitinated CHIP with the mito.*

Response:

We apologize for not clearly describing our data. Our conclusion is that part of monoubiquitinated CHIP is located on mitochondria. In Figure 6i, we evaluated monoubiquitinated CHIP level in mitochondrial extract and that out of mitochondria. The results indicated that only monoubiquitinated CHIP located on mitochondria, which also proved by Figure 6h. To make our conclusion clear, we change the description as “Interestingly, subcellular fractionation studies revealed the only monoubiquitinated CHIP is located on mitochondria (Fig. 6i).”

3. *Fig 7k: I do not understand what the authors mean by “We found that CHIP was preferred but not only to select K48-linked polyubiquitin chains by Parkin”. The immunoblot seems to suggest no difference between HA-K48 and HA-K48R, but most importantly, I do not understand the rationale for this experiment. How would the authors interpret the data if there was a difference or if there was no difference between HA-K48 and HA-K48R? If there is no strong rationale for this experiment, the authors should consider not including the data in the manuscript.*

Response:

We agree with the reviewer’s concerns and comments on ubiquitination subpopulation assay. Following these suggestions, we delete this part of complicated data, which are not associated with the final conclusion tightly.

4. *Check lines 330-332 for figure call-outs.*

Response:

We thank the reviewer for this remind, we have corrected this typo.

5. *Fig 1e: The authors should explain how they interpret LPS-inducible loss of Drp1 phosphorylation.*

Response:

This is an excellent question. As our conclusion, LPS reduces FAM73b expression and triggers mitochondrial fission. Previous research further indicates that mitochondrial fission reduces Ca^{2+} uptake and intramitochondrial Ca^{2+} diffusion¹. Cytosolic Ca^{2+} rise activates the cytosolic phosphatase calcineurin that normally interacts with Drp1. Calcineurin dephosphorylates Drp1 and regulates its translocation to mitochondria². Therefore, all these data implied that Drp1 dephosphorylation may due to the enhanced calcineurin activity. Following the reviewer's suggestion, we add some discussion for this observation in text.

Reviewer #2:

1. *In line 268, the authors wrote that they have examined activation of MAPKs, NF-kB and STAT3. In figure 5, I do not see the analysis of STAT3 activation.*

Response:

We apologize for the error. Actually, STAT3 phosphorylation is not associated with the final conclusion tightly, so we deleted STAT3 data which was shown in the first section. We thank the reviewer for this remind, we have corrected this typo.

2. *I guess that in figure 7, panel "B" should be "C" and "C" should be "B" to be in the right order according to the text.*

Response:

We thank the reviewer for this remind, we have corrected this typo.

3. *Line 331, "(Fig. 7b)" is lacking.*

Response:

We have corrected this typo.

4. *Moreover, again, some typos remain in the text and the English should be also edited.*

Response:

To improve the readability of our manuscript, we used "Nature Publishing Group Language Editing" Golden service to make our manuscript clear and well written.

5. *In figure 2J, the values in the different histograms are not clear.*

Response:

6. Line 344-345, the authors did use HEK293 or HEK293T?

Response:

We apologize for the error. We only use HEK293T cells for all transfect experiments and we have made the correction.

Reference:

1. Szabadkai G, Simoni AM, Chami M, Wieckowski MR, Youle RJ, Rizzuto R. Drp-1-Dependent Division of the Mitochondrial Network Blocks Intraorganellar Ca²⁺ Waves and Protects against Ca²⁺-Mediated Apoptosis. *Molecular Cell* 2004, **16**(1): 59-68.
2. Cereghetti GM, Stangherlin A, de Brito OM, Chang CR, Blackstone C, Bernardi P, *et al.* Dephosphorylation by calcineurin regulates translocation of Drp1 to mitochondria. *Proceedings of the National Academy of Sciences* 2008, **105**(41): 15803-15808.

REVIEWERS' COMMENTS:

Reviewer #1 (Remarks to the Author):

The authors have addressed all of my concerns.

Reviewer #2 (Remarks to the Author):

This revised version is suitable for publication as the concerns raised by the referees have been addressed.